# SeedLoRA: A Fusion Approach to Efficient LLM Fine-Tuning

## Abstract

Despite Low-Rank Adaptation (LoRA)'s popularity for fine-tuning large models, it often exhibits a noticeable performance gap compared to full fine-tuning, particularly in complex tasks such as mathematical reasoning and code generation. Motivated by this discrepancy, we propose a novel fusion approach for LoRA fine-tuned models. Our key insight is that LoRA models trained with different random seeds on the same task often exhibit complementary strengths. In contrast to existing research that typically focuses on fusing models trained on diverse tasks, we explore the potential of combining multiple LoRA models fine-tuned on the same task with different random seeds. This intra-task fusion method aims to leverage the strengths of various fine-tuned models to create a more robust and effective adaptation. To validate our approach, we conducted comprehensive experiments across three key areas: mathematical reasoning, code generation, and general instruction-tuning tasks. The results demonstrate that our fusion method significantly enhances LoRA's performance, outperforming both standalone LoRA models and current fusion methods. Notably, this advancement substantially narrows the gap between LoRA and full fine-tuning, thus offering a more effective approach to model adaptation without the GPU memory burden of full parameter fine-tuning.

## 1 Introduction

Parameter-Efficient Fine-Tuning (PEFT) methods have emerged as promising training schemes in fine-tuning large language models (LLMs), offering a balance between performance and efficiency. Among these, LoRA (Hu et al., 2022) has gained popularity due to its effectiveness and simplicity. Despite its advantages, LoRA often exhibits anoticeable performance gap compared to full fine-tuning approaches, limiting its applicability in scenarios requiring state-of-the-art performance.

Researchers have proposed various approaches to narrow the performance gap between LoRA and full fine-tuning in LLMs. These methods typically fall into three categories: increasing LoRA's capacity, optimizing LoRA's structure, and combining multiple LoRA adaptations. For instance, ReLoRA (Lialin et al., 2024) proposes periodically increasing the rank during training, while DoRA (yang Liu et al., 2024) and MiLoRA (Wang et al., 2024a) suggest alternative low-rank structures and initialization strategies. Techniques such as MultiLoRA (Wang et al., 2023a) and MoLoRA (Zadouri et al., 2024) attempt to leverage multiple LoRA modules, inspired by Mixture of Experts models. While these approaches have shown improvements, they often come at the cost of increased computational complexity or fail to fully close the gap with full fine-tuning, particularly in challenging domains like mathematical reasoning and code generation.

In our investigation of these limitations, we made a key observation: models trained on identical tasks with different random seeds exhibit similar overall performance, yet demonstrate varying proficiency across different subdomains of the task. This opens the opportunity to combine their strengths into a more robust model.

Inspired by this insight, we naturally turn to model merging techniques, which have gained significant attention in the field of LLMs as means to combine knowledge from multiple models without increasing inference costs. However, we find that applying existing merging methods to our scenario presents unique challenges. Specifically, most existing work on model merging focuses on

multi-task scenarios, aiming to integrate capabilities from models trained on different tasks (Wortsman et al., 2022; Ilharco et al., 2022). In Contrast, our experiments reveal that the challenges faced in single-task model merging—our focus—differ substantially from those in multi-task scenarios. To elucidate this distinction, our analysis of cosine similarities reveals a crucial difference: models trained on different tasks exhibit near-zero similarity, indicating orthogonality, which leads to interference issues in multi-task merging. Conversely, models trained on the same task with different seeds show high cosine similarity, suggesting a high degree of shared information. This fundamental difference shifts the primary challenge in single-task merging from interference mitigation to effective information combination and redundancy elimination, necessitating a new approach tailored specifically to single-task model merging.

Building on these insights, we propose SeedLoRA, a novel approach to address the unique challenges of single-task model merging. Our approach capitalizes on the high cosine similarity and shared information between models trained on the same task with different seeds, focusing on effective information combination and redundancy elimination. At the core of SeedLoRA is the Weight Distribution Match technique, which consists of three key steps: (1) analyzing the weight distributions of individual seed-specific models to capture subtle variations from different initializations, (2) performing an initial merge through weighted averaging, leveraging the high similarity to combine complementary strengths, and (3) applying distribution matching to preserve desirable statistical properties. This method aims to create a merged model maintains the effective characteristics of individual models. This method aims to create a merged LoRA model that synergies individual strengths while preserving beneficial properties, potentially narrowing the performance gap with full fine-tuning while maintaining LoRA's computational efficiency.

Our experimental results demonstrate the effectiveness of SeedLoRA. By merging multiple LoRA models with a rank of 8, we achieve performance comparable to full fine-tuning in challenging tasks such as mathematical reasoning and code generation. This approach not only narrows the performance gap between LoRA and full fine-tuning but also preserves the efficiency advantages of PEFT methods.

The main contributions of this paper are:

- A comprehensive analysis of the performance characteristics of LoRA models trained with different seeds on the same task, revealing their complementary strengths in various subdomains.

- Insights into the fundamental differences between single-task and multi-task model merging, highlighting the need for specialized approaches in each scenario.

- The introduction of SeedLoRA, a novel intra-task model merging method that effectively combines information from multiple models while eliminating redundancy.

- Extensive empirical evidence demonstrating the effectiveness of SeedLoRA in narrowing the performance gap between LoRA and full fine-tuning, particularly in complex tasks like mathematical reasoning and code generation.

## 2 PRELIMINARIES

### 2.1 LORA

LoRA has emerged as a key method for efficient fine-tuning of LLMs. It works by injecting trainable rank decomposition matrices into the layers of a pre-trained model, updating only these low-rank matrices during fine-tuning while keeping the original weights frozen. Formally, for a pre-trained weight matrix $W \in \mathbb{R}^{d \times k}$, LoRA introduces the update $W' = W + BA$, where $B \in \mathbb{R}^{d \times r}$ and $A \in \mathbb{R}^{r \times k}$ are low-rank matrices with rank $r \ll \min(d, k)$. Recent research has expanded upon the LoRA framework, exploring various enhancement. These include novel initialization method of $A$ and $B$ matrices in LoRA (MiLoRA (Wang et al., 2024a), Pissa (Meng et al., 2024a), LoRA-GA (Wang et al., 2024b)), higher-rank approaches (MoRA (Jiang et al., 2024), PeriodicLoRA (Meng et al., 2024b), ReLoRA (Lialin et al., 2024), COLA (Xia et al., 2024)), innovative structural modification(DoRA (yang Liu et al., 2024)), and advanced training strategy (LoRA+ (Hayou et al., 2024)).

In addition to these individual enhancements, a significant direction in LoRA research has been the development of Mixture of LoRA techniques. Drawing inspiration from the Mixture of Experts (MoE) paradigm, this approach dynamically combines multiple LoRAs, each potentially specialized for different tasks or domains. Examples include MultiLoRA (Wang et al., 2023a), MoLoRA (Zadouri et al., 2024), LoRAHub (Huang et al., 2023), and HydraLoRA (Tian et al., 2024). By leveraging both LoRA's parameter efficiency and the adaptive capacity of expert models, Mixture of LoRA aims to create more versatile models that can perform effectively across a board range of tasks than single-adaptation LoRA implementations.

## 2.2 MODEL MERGE

Model merging aims to combine the knowledge encoded in multiple trained models into a single, enhanced model. Unlike ensemble methods, which require running multiple models, merged models aim to distill collective knowledge into a single set of parameters, improving efficiency, adaptability, and generalization capacity. Current research in model merging focuses on two main areas: Multi-task Merging and Same/Similar-task Merging. Multi-task merging combines models trained on different tasks into a single model capable of performing multiple tasks, leveraging task-specific knowledge, and maintaining efficiency. Same/Similar-task merging, though less explored, focuses on combining models trained on identical or closely related tasks to enhance robustness and generalization, with studies showing improved performance on shifted data distributions. Most work in this area has been conducted in computer vision, leaving significant opportunities for application in fields like natural language processing. Although a range of methods (Ilharco et al., 2022; Lu et al., 2024; Verma & Elbayad, 2024; Huang et al., 2024; Salamanca et al.; Tam et al.; Deep et al., 2024) for model merging have been proposed, this paper focuses on a selected set of methods that provide distinct ways of combining model parameters. We primarily examine and compare the following merging methods:

- **Model Soup (Wortsman et al., 2022)**: Model soup improves the accuracy of fine-tuned models by averaging the weights of multiple models fine-tuned with different hyperparameters, rather than selecting only the best individual model. This method often outperforms the best individual model on both in-distribution and out-of-distribution data. Model soup demonstrates why we can merge different models and get a better performance.

- **TIES (Yadav et al., 2023)**: TIES-MERGING reduces interference by addressing redundant parameter values and sign disagreements across models through a three-step process: (1) trimming parameters that changed minimally during fine-tuning, (2) resolving sign conflicts across models, and (3) merging only parameters aligned with the agreed-upon sign. This method consistently outperforms other merging techniques across various domains.

- **DARE (Yu et al., 2024)** : DARE drops a large portion of delta parameters and rescales the remaining ones, maintaining performance while reducing redundancy in fine-tuning. Applied before merging, DARE mitigates parameter interference and improves overall performance over individual source model.

## 3 OUR METHOD

### 3.1 MOTIVATION

**Performance Gap between LoRA and Full Fine-Tuning.** Prior research (Biderman et al., 2024) has demonstrated that LoRA usually shows a performance gap compared with full fine-tuning, potentially limiting its application across various tasks. While increasing the rank value and extending training epochs can improve the performance, a notable gap persists, particularly in Math problems and code generation tasks. In Table 1, we conducted the experiments fine-tuning LLaMA2-7B with LoRA from rank=8 to rank=64 on math (MetaMathQA) and code generation (CodeFeedback) tasks (Yu et al., 2023). The results reveal that although increasing the rank value of LoRA from 8 to 64 improves performance on GSM8K and MATH from $39.64$ to $41.05$, a significant gap remains when compared with Full Fine-Tuning on rank 64. A similar trend was observed in code generation tasks, where increasing the rank initially improves performance, but at extremely high ranks, such as 64, performance begins to decline. This indicates that while higher ranks can lead to gains, LoRA still struggles to fully match the performance of full fine-tuning, particularly in complex domains

such as mathematical reasoning and code generation. These observations motivate the need for a novel training strategy and optimization method to further narrow the performance gap, enhancing LoRA's applicability across a wider range of tasks.

| Task | rank=8 | rank=16 | rank=24 | rank=32 | rank=64 | Full FT |
|------|--------|---------|---------|---------|---------|---------|
| **GSM8K** | 64.0 | 65.6 | 64.9 | 64.7 | 65.6 | 66.5 |
| **MATH** | 15.3 | 15.3 | 16.3 | 16.6 | 16.5 | 19.8 |
| **Average** | 39.6 | 40.4 | 40.6 | 40.7 | 41.1 | 43.2 |

Table 1: Fine-Tuning LLaMA-2-7B model with LoRA on MetaMathQA (seed=11).

## 3.2 NARROWING THE PERFORMANCE GAP THROUGH MODEL MERGING

**Analyzing LoRA and Full Fine-Tuning Performance.** We conducted a comprehensive analysis to better understand the performance discrepancy between LoRA and full fine-tuning across various subdomains. Our approach involves visualizing the performance of multiple models trained with different random seeds using both LoRA and full fine-tuning techniques. Specifically, we leverage the Massive Multitask Language Understanding (MMLU) benchmark, which covers a wide range of subjects and allows for fine-grained performance analysis. Figure 1 illustrates the performance of LoRA and full fine-tuning models across different MMLU subdomains. Our findings reveal an interesting pattern: while LoRA models generally underperform compared to full fine-tuning, they exhibit competitive

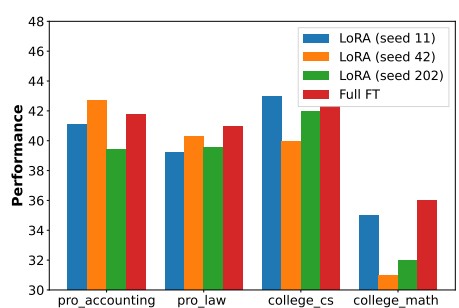

Figure 1: Performance comparison of LoRA and Full FT across MMLU subdomains.

performance in specific subdomains. This nuanced performance distribution led to a key observation: different LoRA models, each trained with unique random seeds, tend to excel in distinct subdomains. Building on this insight, we formulated a promising hypothesis: by strategically merging multiple LoRA models, each with its own specialized strengths, we could potentially achieve performance comparable to full fine-tuning models.

**The Definition of Single-Task model Merging.** To explore our hypothesis of combining LoRA models with diverse strengths, we turn to the concept of model merging, which is the process of combining multiple models to enhance overall performance. While model merging is typically applied to integrate models trained on different tasks, we propose a novel application: merging models trained on a single task with different random seeds to achieve superior performance within that task. We formally define our single-task model merging method as follows:

Let $\theta_{pre}$ be the pre-trained base model, and $\{\theta_1, \theta_2, \ldots, \theta_n\}$ be a set of $n$ models fine-tuned on the same task using LoRA, each with a different random seed $s_i$. Each fine-tuned model $\theta_i$ can be represented as $\theta_i = \theta_{pre} + \tau_i$, where $\tau_i$ is the LoRA delta model for the $i$-th fine-tuned model, obtained using seed $s_i$. Our merging process focuses on these seed-specific delta models:

$$\tau_m = \text{Merge}(\tau_1, \tau_2, \ldots, \tau_n) \tag{1}$$

The merging is performed layer-wise for each LoRA adapter:

$$\tau_m^{(j)} = \text{Merge}(\tau_1^{(j)}, \tau_2^{(j)}, \ldots, \tau_n^{(j)}) \tag{2}$$

where $\tau_i^{(j)}$ represents the $j$-th layer of the $i$-th delta model trained with seed $s_i$. The final merged model is obtained by:

$$\theta_m = \theta_{pre} + \tau_m = \theta_{pre} + \text{Merge}(\tau_1, \tau_2, \ldots, \tau_n) \tag{3}$$

This approach leverages the diverse strengths of multiple LoRA models, each potentially excelling in different subdomains, to create a merged model that approaches or even surpasses the performance of full fine-tuning models. By focusing on models trained on a single task with different seeds, we capture a broader spectrum of task-specific knowledge while maintaining LoRA's efficiency advantages.

### 3.3 THE DIFFERENCE BETWEEN MULTI-TASK AND SINGLE-TASK MODEL MERGING

Before delving into our proposed single-task model merge method, we conducted a cosine similarity analysis to highlight its crucial differences from traditional multi-task approaches. Figure 2 illustrates our findings, revealing a stark contrast between multi-task and single-task scenarios.

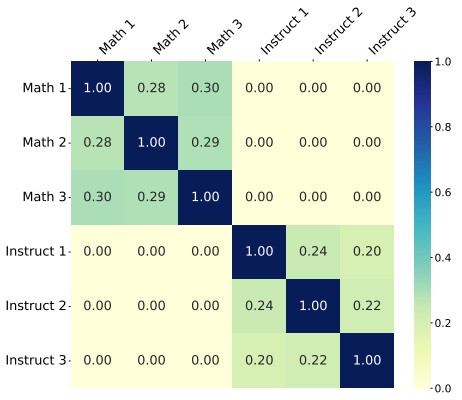

- **Models from Different Tasks (Multi-task Scenario):** The cosine similarity between delta models from different tasks is approximately zero, indicating near orthogonality.

- **Models from the Same Task (Single-task Scenario):** Delta models trained on the same task exhibit cosine similarities consistently greater than zero, suggesting a strong inter-model relationship.

Figure 2: Cosine similarity comparison between models trained on the same and different tasks (MetaMathQA and TULU-v2).

These similarity differences significantly impact merging strategies. In multi-task merging, near-orthogonality leads to interference issues, where preserving task-specific knowledge without degrading performance on other tasks is the primary challenge. Conversely, Single-task merging face the complexity of identifying and leveraging complementary information among largely similar models, albeit with reduced interference risks.

The fundamental differences explain why current multi-task model merging techniques are suboptimal for single-task scenarios. Multi-task methods, such as TIES, are designed to address interference between orthogonal models, which is less relevant in single-task merging. Additionally, techniques that focus on preserving large-magnitude weights to combat interference may not effectively capture the nuanced differences between similar models trained on the same task.

These limitations highlight the need for a novel approach tailored to the unique characteristics of single-task model merging. An effective method for single-task LoRA model merging should leverage the inherent similarities between models while exploiting their subtle differences. This approach aims to surpass the performance of individual LoRA models and potentially approach or exceed the capabilities of full fine-tuning.

### 3.4 SEEDLORA: A DISTRIBUTION MATCHING APPROACH FOR MODEL MERGING

While our previous analysis highlighted the differences between multi-task and single-task model merging, several challenges specific to single-task merging need to be addressed:

- **Information Consolidation:** Models trained on identical tasks with different random seeds often capture overlapping information. Naive merging approaches risk inefficient consolidation, potentially diluting the unique strengths each model has developed for specific aspects of the task.

- **Distribution Dynamics:** The Merging process can alter the overall weight distribution of the resulting model. This shift may lead to unanticipated behavioral changes in the composite model, necessitating considerations of how individuals model contribute to the final merged model

- **Performance Balancing:** Determining the optimal weighting for individual model contribution in the merged model presents siginificant challenges, especially when models perform differently

across various aspects of the task.The merging strategy must carefully balance these contributions to preserve and potentially enhance overall performance.

These challenges necessitate a more sophisticated approach to creating a cohesive merged model with desirable statistical properties. To address this, we introduce Weight Distribution Match based Model Merge. Our method combines the strengths of multiple models trained on the same task while preserving a crucial statistical property: the weight distribution of the individual models. We explain our approach as follows:

1. **Step 1: Individual Model Training and Analysis:** We can individual training multiple model on the same task and then collect multiple models. For each model $\theta_i$, we calculate the mean $\hat{\mu}$ and standard deviation $\hat{\sigma}$ of its weight values.

2. **Step 2: Initial Merging:** Model soup (Wortsman et al., 2022) demonstrates why merging different models can lead to better performance. Therefore, we first create an initial merged model by averaging the weight values from different models in each dimension:

$$\tau_m = \frac{1}{n} \sum_{i=1}^{n} w_i \tau_i \tag{4}$$

where $w_i$ represents the importance weight of the $i$-th model and the default value can be defined as 1, $n$ is the total number of models. This averaging approach is similar to the model soup method, which also combines models through weight averaging.

3. **Step 3: Distribution Matching:** We rescale the values of the merged model to match the mean and standard deviation of a reference model:

$$\tau_m = \hat{\sigma} \cdot \frac{\tau_m - \mu(\tau_m)}{\sigma(\tau_m)} + \hat{\mu} \tag{5}$$

where $\hat{\mu} = \frac{1}{n} \sum_{i=1}^{n} \mu(\tau_i)$ and $\hat{\sigma} = \frac{1}{n} \sum_{i=1}^{n} \sigma(\tau_i)$ are the mean and standard deviation of the reference model, and $\mu(\tau_m)$ and $\sigma(\tau_m)$ are the mean and std of the initially merged weights.

The Weight Distribution Match method effectively addresses the key challenges of single-task model merging. Initial averaging mitigates information redundancy while incorporating diverse information from all models. Subsequent distribution matching preserves model-specific strengths by maintaining statistical properties crucial to individual model performance, particularly in specific task subdomains. This two-step approach not only combines individual model strengths but also tackles distribution shift by rescaling merged weights to match a reference distribution, thus preventing unexpected behaviors in the final model.

## 4 EXPERIMENTAL RESULTS

### 4.1 EXPERIMENTAL SETTING

**Training and Evaluation:** For code generation, we use Code-Feedback (Zheng et al., 2024) as training data, LLaMA2-7B (Touvron et al., 2023) and Mistral-7B-v0.1 (Jiang et al., 2023) serve as base models. We evaluate using HumanEval (Chen et al., 2021), an established benchmark for Python text-to-code generation. For comprehensive assessment, we incorporate HumanEval+ from EvalPlus (Liu et al., 2024). For math reasoning, the MetaMathQA (Yu et al., 2023) dataset is employed to fine-tune on the LLaMA2-7B and Mistral-7B models. The evaluation is conducted using the GSM8k (Cobbe et al., 2021) and MATH (Hendrycks et al., 2021) benchmarks, which are specifically constructed to test the model's capacity for mathematical reasoning and problem-solving. For the general domain, the TÜLU V2 (Wang et al., 2023b) dataset is utilized in training on the LLaMA2-7B (Touvron et al., 2023) and Mistral-7B-v0.1. Following the setting of Open-Instruct (Ivison et al., 2023), we evaluate model on MMLU (Hendrycks et al., 2020), GSM8k (Cobbe et al., 2021), BBH (Suzgun et al., 2022), TyDiQA (Clark et al., 2020), TruthfulQA (Lin et al., 2021) and HumanEval (Austin et al., 2021).

**Implementation Details.** Training is conducted on Nvidia A100 and H100 GPUs using BFloat16 precision. We set weight decay to 0 and employ a cosine learning rate scheduler with a 0.03 ratio linear warmup. For evaluation, we utilize vLLM (Kwon et al., 2023) to conduct our tests, ensuring efficient and scalable inference. More detailed setting is introduced in Table 5.

## 4.2 MATH REASONING

Having delineated our experimental framework, we proceed to present our empirical findings, commencing with an analysis of the performance on math reasoning task. To validate the efficacy of our proposed merge method, we first evaluate the LoRA models with 3 different seeds on GSM8K and MATH, followed by an assessment of our merged model. The experimental results, shown in Table 2, demonstrate that the merged model substantially improve the performance of each independent model. Notably, for the LoRA fine-tuning on LLaMA2-7B, SeedLoRA can improve the performance of vanilla LoRA from 64.1 to 68.6 on GSM8K and from 15.3 to 17.3 on MATH. Furthermore, to evaluate the generalizability of our proposed SeedLoRA, we extend our evaluation to LoRA variants (such as LoRA+ and DoRA) and more advanced pre-trained LLM (such as Mistral-7B). These additional experiments consistently demonstrate performance improvement when using our merged model approach. Finally, we also conduct experiments to compare SeedLoRA with current popular model merge methods, such as Model Soup, TIES and DARE. The experimental results on Table 2 illustrate that these methods can also improve the performance of vanilla LoRA, but SeedLoRA can obtain more performance gain.

| | seed 11 | seed 42 | seed 202 | SeedLoRA | Model Soup | TIES | DARE |
|---|---|---|---|---|---|---|---|
| Evaluating LLaMA2-7B on GSM8K. The performance of Full Fine-Tuning is 66.5. | | | | | | | |
| **LoRA (r=8)** | 64.0 | 63.8 | 64.1 | **68.6** | 66.6 | 65.7 | 65.7 |
| **LoRA+ (r=8)** | 64.4 | 64.7 | 65.4 | **69.8** | 67.0 | 63.2 | 56.7 |
| **DoRA (r=8)** | 64.6 | 64.7 | 64.7 | **68.5** | 66.3 | 66.0 | 67.3 |
| Evaluating LLaMA2-7B on MATH. The performance of Full Fine-Tuning is 19.8. | | | | | | | |
| **LoRA (r=8)** | 15.3 | 15.3 | 14.9 | **17.3** | 15.7 | 16.0 | 15.5 |
| **LoRA+ (r=8)** | 15.4 | 15.5 | 16.0 | **17.4** | 16.1 | 16.7 | 16.4 |
| **DoRA (r=8)** | 15.4 | 15.4 | 14.9 | **17.8** | 16.0 | 15.8 | 15.6 |
| Evaluating Mistral-7B on GSM8K. The performance of Full Fine-Tuning is 78.6. | | | | | | | |
| **LoRA (r=8)** | 75.4 | 75.7 | 76.3 | **80.5** | 79.1 | 75.1 | 75.1 |
| **LoRA+ (r=8)** | 76.5 | 73.5 | 75.9 | **80.3** | 79.7 | 79.4 | 78.7 |
| **DoRA (r=8)** | 77.0 | 75.7 | 76.5 | **80.3** | 77.0 | 79.1 | 78.5 |
| Evaluating Mistral-7B on MATH. The performance of Full Fine-Tuning is 28.5. | | | | | | | |
| **LoRA (r=8)** | 25.9 | 24.8 | 25.4 | **29.0** | 28.5 | 24.8 | 25.0 |
| **LoRA+ (r=8)** | 25.1 | 25.2 | 25.4 | **28.2** | 27.9 | 25.9 | 24.3 |
| **DoRA (r=8)** | 25.9 | 25.3 | 25.8 | **28.7** | 28.3 | 26.5 | 25.7 |

Table 2: Fine-Tuning LLaMA-2-7B and Mistral-7B with LoRA on MetaMathQA.

## 4.3 CODE GENERATION

Building upon our findings in mathematical reasoning, we further evaluate the performance gain of our merged model on the Code Generation task. Table 3 presents the experimental results of individual training models and merged models on CodeFeedback benchmark. The data demonstrates that ourmerged model consistently outperforms individual models in the HumanEval and HumanEval+ tasks. Particularly, SeedLoRA exhibits exceptional performance on HumanEval (+) benchmark, surpassing the best individual LoRA by both 6.1% on LLaMA2-7B and Mistral-7B.

To contextualize our method's performance within a broader range of model merging approaches, we conducted a comparative analysis with popular approaches such as Model Soup, TIES and DARE. Our finding indicates that our method achieves superior performance compared to these existing merge methods. For instance, our method enhance the performance of model soup from 34.1% to 40.2% on vanilla LoRA.

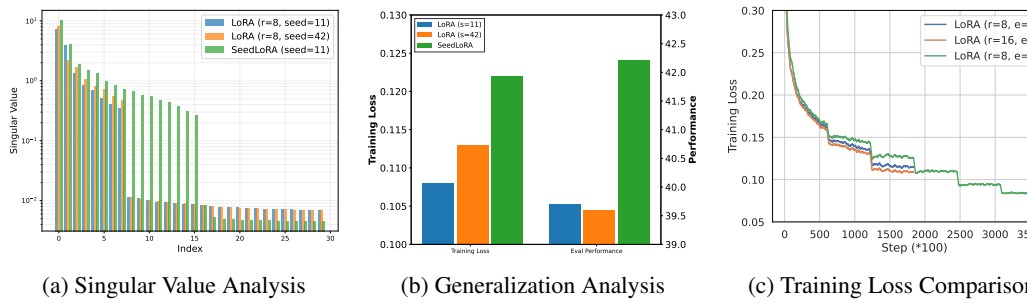

| | (a) Singular Value Analysis | (b) Generalization Analysis | (c) Training Loss Comparison |

Figure 3: (a) Singular Value Analysis. (b) Generalization Analysis (c) Training Loss Analysis.

| | seed 11 | seed 42 | seed 202 | SeedLoRA | Model Soup | TIES | DARE |
|---|---|---|---|---|---|---|---|
| | Evaluating LLaMA2-7B on Humaneval. The performance of Full FT is 40.3. | | | | | | |
| **LoRA (r=8)** | 34.1 | 34.1 | 32.3 | **40.2** | 34.1 | 38.4 | 36.0 |
| **LoRA+ (r=8)** | 36.6 | 35.4 | 32.3 | **39.0** | **39.0** | 32.3 | 30.5 |
| **DoRA (r=8)** | 34.1 | 32.9 | 32.9 | **37.2** | 32.3 | 33.5 | 34.8 |
| | Evaluating LLaMA2-7B on Humaneval+. The performance of Full FT is 37.1. | | | | | | |
| **LoRA (r=8)** | 28.0 | 30.5 | 28.7 | **36.6** | 29.9 | 34.1 | 30.5 |
| **LoRA+ (r=8)** | 31.7 | 34.1 | 29.3 | **36.6** | 34.1 | 28.7 | 27.4 |
| **DoRA (r=8)** | **32.3** | 30.5 | 28.7 | **32.3** | 29.3 | 29.9 | 31.7 |

Table 3: LLaMA2-7B model with LoRA (Delta) on CodeFeedback (Humaneval and Humaneval+).

## 4.4 GENERAL DOMAIN

Having examined the effectiveness of our proposed method SeedLoRA in specialized domains, we now extend our evaluation to general domain instruction tuning tasks. The experimental results, shown in the Table 4, demonstrate that our proposed method continues to improve upon the performance of the best individual model. However, the magnitude of improvement in this domain is less pronounced than observed in math reasoning and code generation. We believe this discrepancy arises from the nature of general domain tasks, where models are required to follow instructions rather than acquire new knowledge, as is often necessary for mathematical and coding tasks. Moreover, this observation underscores the efficacy of our method while also highlighting the challenges of achieving substantial gains in areas where LoRA already performs close to full fine-tuning.

## 4.5 FURTHER DISCUSSIONS

**Why Merge the Models from the Same Task Can Improve the Performance?**

To understand the performance improvements achieved by merging models from the same task, we conduct two key analyses: knowledge fusion and generalization ability.

Firstly, we evaluate whether the merged model can effectively fuse the knowledge from two individual models. We employ Singular Value Decomposition (SVD) to analyze the knowledge representation in each model. Figure 3a illustrates the singular value distribution of individual LoRA models (each with rank 8) and the merged SeedLoRA model. Notably, SeedLoRA exhibits a broader range of non-zero singular values compared to the individual LoRA, suggesting successful knowledge fusion from multiple sources.

Inspired by SWA (Izmailov et al., 2018), which claims that averaging weights can lead to wider optima and better generalization, we investigate whether our model exhibits similar benefits. We analyze the training loss and the performance on downstream evaluation tasks, as shown in Figure 3b and Figure 3c. Interestingly, SeedLORA demonstrates a slightly higher training loss but achieves

|  | MMLU | GSM8K | BBH | TyDiQA | HumanEval | Average |
|---|---|---|---|---|---|---|
| **Full FT** | 48.7 | 31.5 | 42.2 | 51.2 | 21.6 | 39.0 |
| **LoRA (r=8)** | 49.2 | 22.5 | 43.3 | 51.8 | 14.9 | 36.4 |
| **SeedLoRA** | 49.8 | 21.5 | 47.0 | 51.4 | 15.4 | **37.0** |
| **Model Soup** | 50.3 | 20.5 | 45.5 | 50.9 | 15.2 | 36.5 |
| **TIES** | 49.3 | 19.5 | 43.3 | 53.6 | 15.1 | 36.2 |
| **DARE** | 49.6 | 22.5 | 44.3 | 53.4 | 15.3 | 37.0 |
| **LoRA+ (r=8)** | 49.7 | 25.0 | 46.5 | 53.1 | 16.0 | 38.1 |
| **SeedLoRA** | 51.0 | 25.0 | 47.5 | 53.1 | 17.6 | **38.8** |
| **Model Soup** | 51.2 | 24.5 | 45.7 | 52.7 | 17.4 | 38.3 |
| **TIES** | 50.2 | 23.0 | 42.9 | 52.8 | 17.5 | 37.3 |
| **DARE** | 50.1 | 22.0 | 43.1 | 52.9 | 17.4 | 37.1 |
| **DoRA (r=8)** | 49.4 | 25.5 | 46.3 | 50.4 | 16.0 | 37.5 |
| **SeedLoRA** | 50.0 | 29.0 | 46.9 | 52.4 | 15.0 | **38.7** |
| **Model Soup** | 50.4 | 23.0 | 47.7 | 51.0 | 15.2 | 37.5 |
| **TIES** | 49.8 | 22.5 | 45.6 | 53.1 | 14.9 | 37.2 |
| **DARE** | 49.7 | 23.5 | 45.3 | 53.3 | 14.7 | 37.3 |

Table 4: LLaMA-2-7B model with LoRA on Tulu-v2. For the results of LoRA and its variants, we report the best performance of 3 LoRA models, which is trained with different seeds.

superior evaluation performance on downstream evaluation tasks which takes different distributions from the training data. This pattern indicates improved generalization ability, suggesting SeedLoRA learns more robust, task agnostic features rather than overfitting the training data.

These findings on knowledge fusion and generalization provide insight into the mechanisms underlying SeedLoRA's improved performance across various tasks.

**Scaling Results.** To verify the scalability of Seed-LoRA, we conduct experiments on pre-trained models with larger number of parameters. Specially, we evaluate the performance of SeedLoRA on the LLaMA2-13B model, The results are presented in Figure 4. SeedLoRA achieves approximately $2.3\%$ performance gain compared to the best individual LoRA model. This demonstrates that SeedLoRA can effectively improve the performance even on larger pre-trained models, highlighting its scalability and potential for enhance model across different sizes.

**Comparing with Higher Rank LoRA.** To further validate the effectiveness of our approach, we compare SeedLoRA with higher rank LoRA models. This comparison is motivated by the theory about low-rank approximation: $r(\tau_1 + \tau_2) \le r(\tau_1) + r(\tau_2)$,

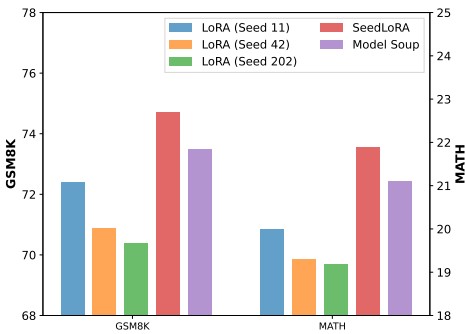

Figure 4: Scaling Results of LLaMA2-13B on MetaMathQA.

where $r(*)$ represents the rank value of a matrix. Since our experiments focus on merging three LoRA models with rank 8, we perform an ablation study comparing our merged model with a single LoRA model with rank 24. As shown in Figure 5, SeedLoRA outperforms the higher-rank LoRA model. This highlights the advantage of SeedLoRA in effectively combining the strengths of multiple lower-rank models, achieving better performance than simply increasing the rank of a single model.

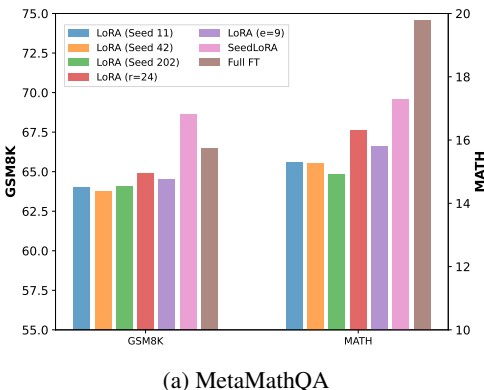 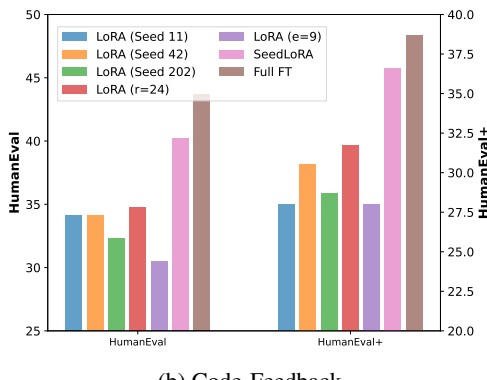

(a) MetaMathQA              (b) Code-Feedback

Figure 5: The comparison between vanilla LoRA training with different seeds, with higher rank, with more epochs and Full Fine-Tuning (Full FT). (a) Comparison on MetaMathQA benchmark. (b) Comparison on Code-Feedback benchmark.

**Training SeedLoRA with Similar Cost as Formal LoRA.** Our method requires obtaining several LoRA models trained on the same tasks. While some suitable models can be found on platforms like Huggingface, it is often necessary to train multiple models by ourselves, potentially incurring additional training time. To address this, we investigate whether we can achieve better performance with comparable training cost using SeedLoRA.

Specifically, we propose an alternative to the standard 3-epoch LoRA fine-tuning of LLaMA2-7B: training 3 individual models with 1 epoch each, then merging these 3 partially trained models. We conduct this experiment on MetaMathQA benchmark, with the results shown in Figure 6. Remarkably, this approach outperforms the standard 3-epoch training while maintaining the same overall training time.

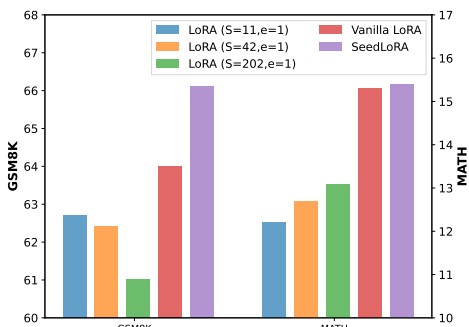

Figure 6: The Performance Comparison between LoRA and Seed LoRA under the similar training cost constraint.

This finding suggests a potentially new, more efficient training paradigm for PEFT.

**Training with More Epochs.** The Training paradigm of SeedLoRA can be regarded as training LoRA with more epochs. To rigorously validate the superior performance of SeedLoRA, we train vanilla LoRA with more epochs and compare our merged model with it. We conduct the experiment on MetaMathQA and Code-Feedback and the comparison result is shown in the Figure 5 and Figure 3c. The results illustrate that SeedLoRA can outperform LoRA training with more epochs on both math reasoning and code generation tasks, although training more epochs can slightly improve its performance.

## 5 CONCLUSIONS

In this paper, we introduced SeedLoRA, a novel single-task model merging approach designed to enhance LoRA fine-tuning. Our method effectively narrows the performance gap between LoRA and full fine-tuning in complex tasks like mathematical reasoning and code generation by combining complementary strengths of models trained with different seeds. Notably, SeedLoRA consistently outperforms existing merging techniques - including Model Soup, TIES, and DARE - in single-task scenarios. The effectiveness of SeedLoRA stems from its ability to fuse knowledge from individual models that specialize in different sub-domains, leading to improved generalization. This approach maintains LoRA's efficiency while achieving comparable performance to full fine-tuning, a finding we demonstrated across various model sizes. By bridging the performance gap between PEFT methods and full fine-tune, our work highlights the potential to enable broader adoption of state-of-the-art LLMs in resource-constrained environments.

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

## A    APPENDIX

### A.1    HYPERPARAMETERS

We propose hyperparameters of three training tasks in Table 5.

| Dataset | Method | $r$ | $\alpha$ | LR | LR Scheduler | Warmup | Epochs | Batch size |
|---|---|---|---|---|---|---|---|---|
| Tülu v2 | FFT | - | - | 2e-5 | cosine | 500 | 2 | 128 |
| | LoRA-like | 8 | 16 | {1e-4,2e-4,3e-4} | cosine | 500 | 2 | 128 |
| MetaMath | FFT | - | - | 2e-5 | cosine | 300 | 3 | 128 |
| | LoRA-like | 8 | 16 | {1e-4,2e-4,3e-4} | cosine | 300 | 3 | 128 |
| Code-Feedback | FFT | - | - | 3e-5 | linear | 300 | 3 | 128 |
| | LoRA-like | 8 | 16 | {1e-4,2e-4,3e-4} | linear | 300 | 3 | 128 |

Table 5: Hyperparameters Setting of fine-tuning on three datasets.

## B    THE PERFORMANCE ANALYSIS ABOUT LORA AND FULL FINE-TUNING ON MBPP

| Task | rank=8 | rank=16 | rank=24 | rank=32 | rank=64 | Full FT |
|---|---|---|---|---|---|---|
| **HumanEval** | 34.1 | 34.1 | 34.8 | 34.8 | 35.4 | 40.3 |
| **HumanEval+** | **28.0** | **32.3** | **31.7** | **31.7** | **31.7** | **37.1** |
| **MBPP (+)** | 45.8 (38.6) | 43.7 (36.0) | 44.2 (36.2) | 46.6 (39.7) | 42.1 (36.2) | 53.1 |
| **Average** | 40.0 (33.3) | 38.9 (34.2) | 39.5 (34.0) | 40.7 (35.7) | 38.8 (34.0) | 46.7 |

Table 6: LLaMA-2-7B model with LoRA (Delta) on Code-Feedback (seed=11).

## C    THE EXPERIMENTAL RESULTS FOR FINE-TUNING MISTRAL-7B ON CODE-FEEDBACK.

| | seed 11 | seed 42 | seed 202 | SeedLoRA | Model Soup | TIES | DARE |
|---|---|---|---|---|---|---|---|
| **LoRA (r=8)** | 53.0 | 51.8 | 48.2 | **57.8** | 53.7 | 55.5 | 56.7 |
| **LoRA+ (r=8)** | 54.3 | 48.8 | 47.6 | **56.7** | 54.3 | 51.8 | 54.3 |
| **DoRA (r=8)** | 54.3 | 55.5 | 45.1 | **56.7** | 54.3 | 56.7 | 55.5 |
| **LoRA (r=8)** | 49.4 | 47.6 | 40.9 | **51.2** | 50.6 | 49.4 | 50.0 |
| **LoRA+ (r=8)** | 48.2 | 43.9 | 40.2 | **49.4** | 48.2 | 47.6 | 48.8 |
| **DoRA (r=8)** | 47.6 | 49.4 | 42.1 | **49.4** | 48.8 | 51.8 | 49.4 |

Table 7: Mistral-7B model with LoRA (Delta) on CodeFeedback (HumanEval and HumanEval+).

# D   THE EXPERIMENTAL RESULTS FOR FINE-TUNING MISTRAL-7B ON TULU-v2

|  | MMLU-0 | GSM8K | BBH | TyDiQA | HumanEval | Average |
|---|---|---|---|---|---|---|
| **LoRA (r=8)** | 59.4 | 46.0 | 54.99 | 59.93 | 33.78 | 50.82 |
|  | 58.8 | 44.0 | 56.57 | 59.54 | 35.51 | 50.88 |
|  | 58.2 | 50.5 | 58.70 | 59.01 | 31.37 | 51.56 |
| **SeedLoRA** | 60.7 | 52.5 | 58.70 | 61.83 | 33.99 | 53.54 |
| **TIES** | 58.3 | 42.5 | 53.79 | 60.35 | 33.74 | 49.73 |
| **DARE** | 58.5 | 42.0 | 56.20 | 60.51 | 35.39 | 50.52 |
| **LoRA+ (r=8)** | 60.8 | 45.0 | 59.44 | 58.23 | 34.08 | 51.51 |
|  | 61.2 | 45.5 | 59.72 | 59.70 | 32.19 | 51.66 |
|  | 60.5 | 47.0 | 58.61 | 59.06 | 32.04 | 51.44 |
| **SeedLoRA** | 61.8 | 47.5 | 61.11 | 59.64 | 34.57 | 52.92 |
| **TIES** | 60.6 | 46.0 | 57.87 | 58.78 | 34.08 | 51.46 |
| **DARE** | 60.4 | 41.0 | 56.29 | 59.24 | 34.23 | 50.23 |
| **DoRA (r=8)** | 61.1 | 46.0 | 58.79 | 58.90 | 34.32 | 51.82 |
|  | 60.3 | 52.0 | 58.79 | 60.09 | 33.10 | 52.85 |
|  | 60.3 | 52.0 | 58.51 | 59.89 | 32.92 | 52.72 |
| **SeedLoRA** | 61.6 | 50.5 | 61.11 | 60.07 | 33.53 | 53.36 |
| **TIES** | 60.5 | 46.5 | 58.24 | 58.24 | 35.30 | 51.75 |
| **DARE** | 60.5 | 44.0 | 57.40 | 59.29 | 35.60 | 51.35 |
| **LoRA (r=24)** | 60.4 | 46.5 | 57.40 | 59.58 | 31.76 | 51.12 |
| **LoRA (epoch=6)** | 56.5 | 47.0 | 53.98 | 55.53 | 32.46 | 49.09 |

Table 8: Mistral-7B model with LoRA (Delta) on Tulu-v2.

# E   WEIGHT DISTRIBUTION ANALYSIS

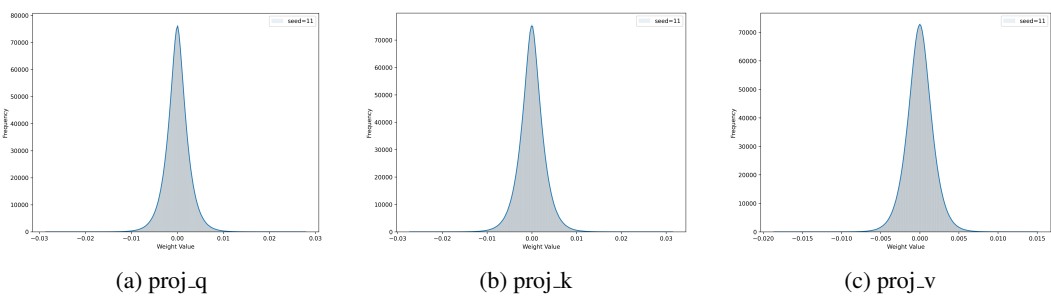

(a) proj_q          (b) proj_k          (c) proj_v

Figure 7: Weight Distribution in Attention-based Layers.

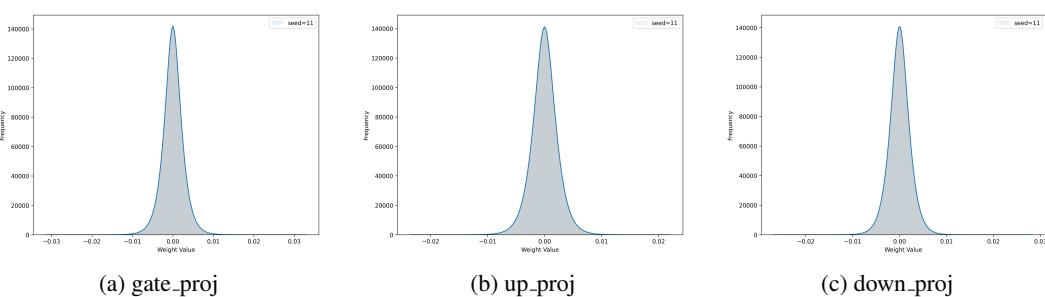

(a) gate_proj  (b) up_proj  (c) down_proj

Figure 8: Weight Distribution in MLP Layers.

