# OpenReview forum: "SeedLoRA: A Fusion Approach to Efficient LLM Fine-Tuning"
_ICLR.cc/2025/Conference — Submitted to ICLR 2025_

### Official Review · Reviewer_qLae · 2024-10-28

**Soundness:** 4
**Presentation:** 3
**Contribution:** 3
**Rating:** 6
**Confidence:** 3

**Summary:**

Parameter-Efficient Fine-tuning (PEFT) is an important direction in fine-tuning large language models (LLMs). The most popular PEFT method is LoRA. LoRA and its variants have exhibited wide performance gap compared to the full fine-tuning in LLM in some complexity tasks, such as mathematical reasoning, and code generation. The key insight brought up by this paper is that some of the limitations in performance can be addressed by exploring merging the models that are trained on the same single task with different random seeds. The core idea of the proposed SeedLoRA is a weight distribution match technique. This technique allows one to create a merged LoRA model that synergies individual strengths while preserving beneficial properties in solving some complexity tasks. This helps to narrow the performance gap with the full fine-tuning in mathematical reasoning and code generation. The proposed approach beats the other existing merging techniques such as Model Soup, TIES< and DARE in the single-task scenarios.

**Strengths:**

This paper introduces a novel approach to model merging. Unlike existing techniques that primarily cater to multi-task merging, the proposed method with random seed focuses on single tasks. This unique approach is particularly suitable for addressing the parameter-efficient fine-tuning problem in mathematical reasoning and code generation tasks. The innovative weight distribution matching further enhances the novelty and utility of the proposed approach.

From an experimental presentation perspective, the paper seems to have made reasonable efforts to demonstrate that the proposed approach really helps with the LLM fine-tuning on the specific LoRA weak tasks such as mathematical reasoning and code generation. The proposed approach has shown stronger results than other interesting and recent methods such as DARE, TIES, and Model Soup. From the results, it is clear that SeedLoRA beats the other standalone LorA and existing multi-task merging techniques on several tasks.

The paper does well in communicating the motivation of the proposed approach. It follows a reasonable progression and is easy to read and understand. I also find it useful to read the specific implementation details and hyperparameter settings, which improve the reproducibility of the paper.

It is important to bridge the performance gap in tasks such as mathematical reasoning and code generation for PEFT for LLMs fine-tuning. Fully fine-tuning the LLMs requires expensive GPU clusters. The previous LoRA and its variants were not good enough in some of the complex but important tasks such as mathematical reasoning and code generation. This gap in performance in mathematical reasoning and code generation fundamentally limits the progress in the field. The typical methods to address this mainly focus on multi-task scenarios, while the proposed approach focuses on single tasks. The proposed SeedLoRA appraoch provides a new perspective to narrow the performance gap in PEFT. In that sense, this paper is significantly interesting to the community.

**Weaknesses:**

There are some issues with the paper as well.

1. Limited impact scope. LoRA and its variants have performed well in multiple tasks and are close to full fine-tuning. The proposed SeedLoRA focuses only on complex tasks such as code generation and mathematical reasoning. The fact that the SeedLoRA only improves the performance on those particular tasks seems quite limited. Would SeedLoRA be helpful in any other tasks outside of these? What are the patterns that the proposed approach would work and that the proposed approach would not work? The experiments in this paper do not provide those answers. That would be very limited if the proposed approach only works on single tasks in code generation or mathematical reasoning.

2. Is SeedLoRA practical? A key motivation behind LoRA is its ability to fine-tune LLMs on relatively inexpensive and limited GPUs quickly. The proposed SeedLoRA, however, would significantly increase the time required to obtain multiple models for a single task, raising concerns about its practicality. The number of models needed for a single task is also unclear, further questioning the feasibility of the approach.

3. The proposed distribution matching algorithm focuses on rescaling the weights by matching the mean and standard deviation, which implies that the model weights have a specific distribution pattern. This seems to be oversimplifying the model weight distribution. It is clear that one could have different options, such as looking into the K-L divergence. Those alternative directions and experiments are missing. It is unclear why the proposed approach makes a technical choice in equation (5).

**Questions:**

1. It is cool that the proposed model merging technique works, but i am wondering if the simple ensemble technique would work the same. Let's say we can train the model with different architecture, and then just use the mixture of the experts, would that beat the proposed approach?

2. It is not clear if the random seeds are impacting only the model weight initialization or also on the data distribution order.

3. What's the computational cost overhead to train the proposed approach? How would that compared against the baseline approaches.

4. Does the proposed approach need to apply the merging on all the weights?

---

> ### Author Response · Authors · 2024-11-24
> **Response to Reviewer qLae**
>
> Thank you for your constructive and positive feedback, we carefully address your concerns below.
>
> ### **W1:**
> >Limited impact scope. LoRA and its variants have performed well in multiple tasks and are close to full fine-tuning. The proposed SeedLoRA focuses only on complex tasks such as code generation and mathematical reasoning. The fact that the SeedLoRA only improves the performance on those particular tasks seems quite limited. Would SeedLoRA be helpful in any other tasks outside of these? What are the patterns that the proposed approach would work and that the proposed approach would not work? The experiments in this paper do not provide those answers. That would be very limited if the proposed approach only works on single tasks in code generation or mathematical reasoning.
>
> Thank you for your constructive comment. We want to respectfully clarify that SeedLoRA's impact extends beyond code generation and mathematical reasoning. Our comprehensive evaluation spans three major areas:
>
> 1.  Mathematics Reasoning
> 2.  Code Generation
> 3.  General Domain Instruction Tuning [1]
>
> Particularly, our evaluation on General Domain Instruction Tuning (as shown in Table 4 and Table 8) demonstrates SeedLoRA's broad applicability across diverse domains:
>
> -   Factual Knowledge (MMLU)
> -   Reasoning Ability (GSM8K, BBH)
> -   Multilingual Capability (TyDiQA)
> -   Coding Skills (HumanEval)
>
> The experimental results show consistent improvements over vanilla LoRA across these domains. For instance, on LLaMA2-7b, SeedLoRA improved the average performance from 36.4 to 37.0, demonstrating its effectiveness not just in specialized tasks but also in general domain applications.
>
> [1] Camels in a Changing Climate: Enhancing LM Adaptation with Tulu 2
>
> ### **W2:**
> >Is SeedLoRA practical? A key motivation behind LoRA is its ability to fine-tune LLMs on relatively inexpensive and limited GPUs quickly. The proposed SeedLoRA, however, would significantly increase the time required to obtain multiple models for a single task, raising concerns about its practicality. The number of models needed for a single task is also unclear, further questioning the feasibility of the approach.
>
> Thank you for this insightful comment. We demonstrate that SeedLoRA remains practical while offering significant performance benefits:
>
> 1.  **Optimal Number of Models for Merging** We conducted extensive experiments varying the number of merged models from 2 to 5 on MetaMathQA. The results show:
>
> Model | Method | Task | number=1 | number=2 | number=3 | number=4 | number=5 |
> |---------|---------|------|--------|--------|--------|--------|--------|
> LLaMA2-7b | LoRA | GSM8K | 64.0 | 68.2 | 68.6 | 68.5 | 68.9
> LLaMA2-7b | LoRA | MATH | 15.3 | 16.2 | 17.3 | 16.9 | 16.5
>
> Our findings reveal:
>
> -   Even merging just 2 models yields substantial improvements: GSM8K accuracy increases from 64.0% to 68.2%, and MATH accuracy from 15.3% to 16.2%
> -   Performance typically plateaus at 3 models, with minimal gains from additional merges. For instance, merging 3 models (68.6%, 17.3%) achieves comparable results to merging 4 (68.5%, 16.9%) or 5 models (68.9%, 16.5%)
> -   For practical applications, merging 2-3 models offers an optimal trade-off between performance and computational cost
>
> 2.  **Efficiency and Resource Requirements** SeedLoRA maintains the key advantages of LoRA while offering enhanced performance:
>
> -   **Memory Efficiency**: SeedLoRA introduces no additional memory overhead compared to standard LoRA, preserving its compatibility with resource-constrained GPUs
> -   **Training Time Optimization**: While training multiple models does increase total computation time, we demonstrate an efficient alternative approach:
>     -   Instead of the standard 3-epoch LoRA fine-tuning, we can train 3 individual models for 1 epoch each and merge them.
>     -   Our experiments on MetaMathQA (Figure 6) show this approach outperforms standard 3-epoch LoRA training while maintaining the same total training duration.
>     -   This demonstrates that SeedLoRA can achieve superior results without necessarily requiring additional training time when properly configured.
>
> These findings demonstrate that SeedLoRA provides a practical enhancement to LoRA, offering improved performance while maintaining its core benefits of efficient fine-tuning on limited hardware.

---

> > ### Author Response · Authors · 2024-11-24
> > **Response to Reviewer qLae - Continue**
> >
> > ### **W3:**
> > >The proposed distribution matching algorithm focuses on rescaling the weights by matching the mean and standard deviation, which implies that the model weights have a specific distribution pattern. This seems to be oversimplifying the model weight distribution. It is clear that one could have different options, such as looking into the K-L divergence. Those alternative directions and experiments are missing. It is unclear why the proposed approach makes a technical choice in equation (5).
> >
> > Thank you for these insightful comments about our distribution matching approach. While we acknowledge that mean and standard deviation matching may appear simple, our choice was guided by empirical observations and analysis:
> >
> > 1.  Weight Distribution Characteristics: Through empirical analysis, we observed that the weight distributions of LoRA models trained on the same task consistently exhibit Gaussian-like characteristics, with most weights concentrated near zero. This observation is particularly relevant because:
> >     - For approximately Gaussian distributions, mean and standard deviation are sufficient statistics
> >     - This pattern holds consistently across different random seeds and training runs
> > 2.  Computational Efficiency: While more complex distribution matching methods like KL divergence could potentially capture finer distribution details, our approach offers an effective balance between performance and computational overhead. This is particularly important given LoRA's core goal of efficient adaptation.
> > 3.  Robustness and Stability: Our method provides a reliable way to preserve the essential statistical properties of the weights while avoiding potential instabilities that could arise from matching higher-order moments or using more complex distribution metrics.
> >
> > ### **Q1:**
> > >It is cool that the proposed model merging technique works, but i am wondering if the simple ensemble technique would work the same. Let's say we can train the model with different architecture, and then just use the mixture of the experts, would that beat the proposed approach?
> >
> > Thank you for this constructive comment. While ensemble techniques are indeed powerful for boosting model performance, their application to Large Language Models (LLMs) presents unique challenges. Unlike traditional classification tasks with fixed output classes, LLMs must generate sequences of tokens from a vast vocabulary space, making simple averaging or voting-based ensemble strategies less straightforward.
> >
> > Moreover, the memory requirements for ensemble or Mixture-of-Experts (MoE) approaches would be substantially higher, as they require loading multiple complete models into GPU memory during inference. For example, a simple ensemble of N models would multiply the memory footprint by N times, making it impractical for deployment in resource-constrained environments. In contrast, our proposed model merging technique maintains a single model's memory footprint while capturing the diverse strengths of multiple models.
> >
> > ### **Q2:**
> > >It is not clear if the random seeds are impacting only the model weight initialization or also on the data distribution order.
> >
> > Thank you for this insightful comment regarding the random seed effects. To address this concern, we conducted additional experiments to disentangle the impact of random seeds on weight initialization versus data ordering. We investigated two controlled settings:
> >
> > 1.  Fixed weight initialization with varied data ordering (using different random seeds for data shuffling)
> > 2.  Fixed data ordering with varied weight initialization (using different random seeds for initialization)
> >
> > Our experimental results on MetaMathQA using LLaMA2-7b (averaged across 3 different seeds) are summarized below:
> >
> > Method | GSM8K | MATH |
> > |---------|---------|------|
> > fixed weight initialization, varied data ordering | 68.8 | 17.1
> > fixed data ordering, varied weight initialization | 68.3 | 17.2
> > vanilla LoRA | 64.0 | 15.3
> > SeedLoRA | 68.6 | 17.3
> >
> > The results demonstrate that both controlled conditions yield substantial improvements over the vanilla LoRA baseline. Specifically, varying either the weight initialization or the data ordering alone leads to performance gains of approximately 4-5 percentage points on GSM8K and 2 percentage points on MATH. This suggests that both factors contribute to the model's enhanced performance, with neither dominating the improvement.

---

> > > ### Author Response · Authors · 2024-11-24
> > > **Response to Reviewer qLae - Continue**
> > >
> > > ### **Q3:**
> > > >What's the computational cost overhead to train the proposed approach? How would that compared against the baseline approaches.
> > >
> > > Thank you for this constructive comment. Regarding computational costs, SeedLoRA does not introduce additional memory overhead compared to baseline LoRA methods. In terms of training time, our approach typically requires 2-3x the vanilla LoRA training time, as we need to train and merge 2-3 models based on our analysis.
> > >
> > > However, we also demonstrate in Section 4.5 and Figure 6 that SeedLoRA can achieve superior performance even when constrained to similar computational costs as standard LoRA. Specifically, instead of the conventional 3-epoch LoRA fine-tuning of LLaMA2-7B, we explored an alternative approach: training three individual models for 1 epoch each and then merging them. Our experiments on MetaMathQA show that this computationally-equivalent version of SeedLoRA (merging three 1-epoch models) outperforms the standard 3-epoch training while maintaining the same total training time.
> > >
> > > ### **Q4:**
> > > >Does the proposed approach need to apply the merging on all the weights?
> > >
> > > Thank you for this insightful question. While our original experiments merged all updated parameters, we conducted additional experiments to investigate the impact of merging different subsets of layers, specifically analyzing the contributions of attention-related layers versus feed-forward network (FFN) layers.
> > >
> > > In our setup with LoRA fine-tuning of 7 layers per block, we compared merging:
> > >
> > > 1.  Attention-related layers only (q_proj, k_proj, v_proj)
> > > 2.  FFN-related layers only (o_proj, gate_proj, up_proj, down_proj)
> > >
> > > Our experimental results on LLaMA2-7b demonstrate that FFN layers are the primary contributors to performance improvements:
> > >
> > > Method | GSM8K | MATH |
> > > |---------|---------|------|
> > > vanilla LoRA (no merge) | 64.0 | 15.3
> > > Attention-related layers | 64.1 | 15.3 |
> > > FFN related layers | 68.5 | 17.1 |
> > > SeedLoRA (merge both layers)| 68.6 | 17.3
> > >
> > > Notably, merging only FFN-related layers achieves comparable performance to full parameter merging (SeedLoRA), while merging attention-related layers alone shows minimal improvement over the baseline. These findings suggest that the benefits of our approach primarily stem from the FFN components. We will incorporate these results and their implications in our revised version.

---

> > ### Comment · Reviewer_qLae · 2024-11-25
> >
> > Thanks for providing detailed answers.
> >
> > 1. I think it makes sense to provide additional experiments to validate the point by looking into the merging model experiment.
> >
> > 2. The technique merging multiple models seems to be related to the idea in model soup paper, which showed that averaging model parameters can boost the model performance. Is this the trick that is used here? If so, please properly cite and discuss the related works in the updated version of the paper.

---

> > > ### Author Response · Authors · 2024-11-26
> > > **Response to Reviewer qLae**
> > >
> > > Thanks for your constructive and inspiring feedback, we carefully address your concerns below.
> > >
> > > >1.  I think it makes sense to provide additional experiments to validate the point by looking into the merging model experiment.
> > >
> > > Thank you for your constructive comment. We acknowledge that additional experiments would further validate SeedLoRA's performance in model merging. To address your concern, we have conducted additional experiments on commonsense tasks [1,2], following the experimental settings of DoRA [2] on LLaMA2-7b.
> > >
> > > Method |seed |     BoolQ | PIQA | SIQA | HellaSwag | WinoGrande | ARC-e | ARC-c | OBQA | Average
> > > |---------|---------|------|--------|-------|-------|--------|---------|-------|--------|----------------|
> > > LoRA | 42 | 68.5 | 81.2 | 78.2 | 82.0 | 81.5 | 80.8  | 66.6 | 82.0 | 77.6
> > > LoRA | 202 | 68.9 | 80.9 | 78.9 | 86.5 | 81.5 | 79.5  | 64.3 | 80.4 | 77.6
> > > SeedLoRA | / | **71.9** | **83.2** | **80.4** | **90.7** | **84.1** |  **83.5** | **69.7** | **83.6** | **80.9 ($\uparrow$ 3.3)**
> > >
> > > The results demonstrate that SeedLoRA consistently outperforms individual LoRA models across all evaluated metrics through merging just two models. We are also planning to conduct experiments with LoRA fine-tuning on quantized LLMs, specifically using QLoRA [3] and LoftQ [4], and will provide these results in the coming days. Furthermore, we welcome the opportunity to conduct additional experiments to verify SeedLoRA's performance in the final version. If you have any specific datasets or tasks of interest, we would be glad to include them in our evaluation.
> > >
> > > [1] LLM-Adapters: An Adapter Family for Parameter-Efficient Fine-Tuning of Large Language Models.
> > >
> > > [2] DoRA: Weight-Decomposed Low-Rank Adaptation, ICML 2024
> > >
> > > [3] QLoRA: Efficient Finetuning of Quantized LLMs, NeurIPS 2023.
> > >
> > > [4] Loftq: Lora-fine-tuning-aware quantization for large language models, ICLR 2024.
> > >
> > > >2. The technique merging multiple models seems to be related to the idea in model soup paper, which showed that averaging model parameters can boost the model performance. Is this the trick that is used here? If so, please properly cite and discuss the related works in the updated version of the paper.
> > >
> > > Thank you for your constructive comment. The model soup approach demonstrates how averaging weights across multiple models can enhance performance on both in-distribution and out-of-distribution data.
> > >
> > > We have addressed this connection in two ways in our revision: First, we provide a brief overview of model soup's key concepts in Section 2.2, along with the appropriate citation. Second, we acknowledge in Section 3.4 that our method's second step shares similarities with model soup in its use of weight averaging, and we discuss how model soup's insights help explain the improved performance we observe in our merged models.

---

> > > > ### Author Response · Authors · 2024-12-03
> > > > **Response to Reviewer qLae - Continue**
> > > >
> > > > Thank you very much for your constructive comment!
> > > >
> > > > To further address concerns regarding additional experimental validation, we conducted experiments on fine-tuning quantized pre-trained models. Specifically, we utilized QLoRA and LoftQ methods to fine-tune LLaMA2-7B on the GSM8K dataset and analyzed the performance of SeedLoRA. The results, presented in the table, demonstrate that SeedLoRA consistently improves performance: accuracy increased from 39.0% to 42.2% when using QLoRA, and from 39.0% to 42.6% with LoftQ.
> > > >
> > > > |Model | Method |   seed=11   |  seed=42  | seed=202 | SeedLoRA |
> > > > |---------|------|-------|---------|-------|---------|
> > > > LLaMA-7b | LoftQ (4bit) |  38.2  |  39.0  |  38.2  | **42.6** |
> > > > LLaMA-7b | QLoRA (4bit) | 38.2  |  37.7  |  39.0  |  **42.2**  |

---

### Official Review · Reviewer_tN6o · 2024-11-03

**Soundness:** 2
**Presentation:** 2
**Contribution:** 2
**Rating:** 5
**Confidence:** 4

**Summary:**

This paper proposes a fusion strategy for merging LoRA-tuned Large Language Models. The basic idea is to merge models trained on the same tasks with different random seeds.

**Strengths:**

- Introducing single-task model merging, within the domain of parameter-efficient fine-tuning for LLMs.
- Achieving comparable performance to full fine-tuning on challenging tasks such as math reasoning and code generation.

**Weaknesses:**

- Lack of novelty: The motivation is good. However, the approach for model merging is quite straightforward. It is unclear why such simple averaging or model normalization works. It lacks insights and explanations.

- Lack of clarity. The storyline was good at the beginning. I was confused by the proposed model merging or fusion. Specifically, in Line 290, it is unclear about the reference model. The authors mentioned that it can be the best-performing individual model. How do you know the best model? Model merging is supposed to be done without looking into the test data. If you merge models according to the testing performance, this is wrong!

**Questions:**

- Please answer the issues pointed out in the weakness.

- Lack of the whole picture: What is the limitation of your work? In addition to showing your merged model can achieve the performance of full fine-tuning, you also need to explain what the cost is, when it works and when it doesn't, and why. How many models do you need for merging to reach full fine-tuning performance?

---

> ### Author Response · Authors · 2024-11-24
> **Response to Reviewer tN6o**
>
> Thanks for your constructive comments, we carefully address your concerns below.
>
> ### **W1:**
> >Lack of novelty: The motivation is good. However, the approach for model merging is quite straightforward. It is unclear why such simple averaging or model normalization works. It lacks insights and explanations.
>
> Thanks for your constructive comment. Recent work has demonstrated that parameter values from different training runs can be interpolated without increasing loss due to linear mode connectivity (LMC) [1, 2, 3]. When multiple models are fine-tuned from the same pre-trained model, they share common optimization patterns. This shared foundation means these models can be effectively merged without complex neuron alignment procedures, a property known as permutation symmetry in technical terms.
>
> Our analysis extends these insights by revealing fundamental differences between multi-task and single-task model merging scenarios. Through cosine similarity analysis, we show that models trained on different tasks exhibit near-zero similarity (orthogonality), while models trained on the same task with different seeds demonstrate high cosine similarity, indicating substantial shared information. This finding aligns with weight averaging (WA) literature [4, 5, 6], which shows that under shared pre-training, weights can be linearly interpolated despite architectural non-linearities.
>
> Our experiments demonstrate that LoRA models trained with different seeds exhibit complementary strengths across task subdomains. For instance, our analysis of MMLU subdomains shows that different LoRA models excel in distinct areas - one seed may perform better in professional accounting while another excels in college mathematics. This complementarity, combined with their shared optimization foundation, enables effective merging without complex alignment procedures.
>
> The effectiveness of our Weight Distribution Match technique is validated by empirical results: on GSM8K, SeedLoRA improves performance from 64.0% to 68.6%, surpassing both full fine-tuning (66.5%) and existing merging methods. Our SVD analysis shows broader non-zero singular value distributions in merged models, indicating successful knowledge fusion. This approach enables multiple benefits: enhanced performance on individual tasks (like math problem-solving), improved generalization to new scenarios, and robust performance across different subdomains of the task.
>
>
> [1] WARM: On the Benefits of Weight Averaged Reward Models, ICML 2024.
>
> [2] Linear Mode Connectivity and the Lottery Ticket Hypothesis, ICML 2020.
>
> [3] What is being transferred in transfer learning?, NeurIPS 2020.
>
> [4] Model soups: averaging weights of multiple fine-tuned models improves accuracy without increasing inference time, ICML 2022.
>
> [5] Diverse Weight Averaging for Out-of-Distribution Generalization, NeurIPS 2022.
>
> [6] Model Ratatouille: Recycling Diverse Models for Out-of-Distribution Generalization, ICML 2023.
>
> ### **W2:**
> >Lack of clarity. The storyline was good at the beginning. I was confused by the proposed model merging or fusion. Specifically, in Line 290, it is unclear about the reference model. The authors mentioned that it can be the best-performing individual model. How do you know the best model? Model merging is supposed to be done without looking into the test data. If you merge models according to the testing performance, this is wrong!
>
> We thank the reviewer for this important observation. We would like to clarify that in our current work and experiments, we treat all individual models as equally important and do not perform any selection based on test data performance. The mention of using a 'best-performing model' in Line 290 was intended only as a theoretical possibility for future work, and we agree that this reference may cause confusion. We will remove this statement in the revised version to maintain clarity and avoid any misunderstanding about our methodology. To be explicit, our model merging approach is conducted without any access to or consideration of test data performance. In the revised version, we will update Line 290 to explicitly state that our model merging approach treats all constituent models equally, removing any reference to the selection of a best-performing model.

---

> > ### Author Response · Authors · 2024-11-24
> > **Response to Reviewer tN6o - Continue**
> >
> > ### **Q2:**
> > >Lack of the whole picture: What is the limitation of your work? In addition to showing your merged model can achieve the performance of full fine-tuning, you also need to explain what the cost is, when it works and when it doesn't, and why. How many models do you need for merging to reach full fine-tuning performance?
> >
> > Thank you for this constructive feedback. We address your concerns about limitations, costs, and applicability of our work as follows:
> >
> > 1.  **Limitations and Training Cost**: The primary limitation of our approach is the need for training multiple models (typically 2-3) with different seeds, which increases the total training cost. While this introduces a 2-3x computational overhead compared to vanilla LoRA, we demonstrate in Section 4.5 and Figure 6 that this can be mitigated through an efficient training strategy: instead of training one model for 3 epochs, we train 3 models for 1 epoch each and merge them. Our results show this approach outperforms standard 3-epoch training while maintaining the same total training time. Importantly, we maintain the same memory footprint as vanilla LoRA throughout training.
> > 2.  **Number of Models Required**: We conducted extensive experiments on MetaMathQA to determine the optimal number of models for merging. Our findings show:
> >
> >
> > Model | Method | Task | number=1 | number=2 | number=3 | number=4 | number=5 |
> > |---------|---------|------|--------|--------|--------|--------|--------|
> > LLaMA2-7b | LoRA | GSM8K | 64.0 | 68.2 | 68.6 | 68.5 | 68.9
> > LLaMA2-7b | LoRA | MATH | 15.3 | 16.2 | 17.3 | 16.9 | 16.5
> >
> > -   Merging just 2 models yields significant improvements (GSM8K: 64.0→68.2; MATH: 15.3→16.2)
> > -   Performance typically plateaus at 3 models, with minimal gains from additional models
> > -   Based on these results, we recommend using 3 models for optimal performance, or 2 models when prioritizing training efficiency
> >
> > 3.  **Applicability and Performance Analysis**: SeedLoRA demonstrates broad applicability across various domains including mathematical reasoning, code generation, and general instruction following. However, its effectiveness varies based on:
> >
> > -	Task Difficulty:
> >
> > 	-   For standard tasks like GSM8K, SeedLoRA achieves comparable or better performance than full fine-tuning
> > 	-   On more challenging tasks like MATH, performance depends on the base model's capability
> >
> > - Base Model Strength:
> >
> > 	-   With LLaMA2-7b, while SeedLoRA improves upon standard LoRA, it shows a performance gap compared to full fine-tuning on MATH (17.3 vs 19.8)
> > 	-   With the stronger Mistral-7b-v0.1, SeedLoRA achieves comparable performance to full fine-tuning (29.0 vs 28.5)
> >
> > This analysis suggests that SeedLoRA's effectiveness correlates with both task complexity and base model capability. The performance gap with full fine-tuning is more pronounced when using weaker base models (e.g., LLaMA2-7b) on challenging tasks (e.g., MATH), primarily due to the larger initial performance disparity between LoRA and full fine-tuning in these scenarios.

---

> > ### Comment · Reviewer_tN6o · 2024-11-28
> > **Response to the authors' rebuttal**
> >
> > While I appreciate the authors' substantial efforts in addressing the reviewers' concerns during the rebuttal, I regret to say that I still feel the paper does not meet the bar for ICLR.
> >
> > As Reviewer Eapc also noted, the lack of novelty remains a significant issue. The core contribution appears to rely on model averaging, which, despite being presented in a complex manner, does not introduce substantial methodological innovation. This is evident in the discussions between the authors and Reviewer Eapc.
> >
> > Although the empirical results are impressive, there remains a lack of clarity about the underlying reasons for their success. For instance, as the authors suggest, models with different random seeds perform well on distinct tasks. However, it remains unclear how an averaged model can achieve strong performance across all tasks.
> >
> > Overall, while the experimental findings are interesting, the methodological contribution of this work is limited.

---

> > > ### Author Response · Authors · 2024-12-03
> > > **Response to Reviewer tN6o**
> > >
> > > Thank you for your constructive feedback on our methodology and contributions. We address your core concerns below:
> > >
> > > >The core contribution appears to rely on model averaging, which, despite being presented in a complex manner, does not introduce substantial methodological innovation. This is evident in the discussions between the authors and Reviewer Eapc.
> > > Although the empirical results are impressive, there remains a lack of clarity about the underlying reasons for their success. For instance, as the authors suggest, models with different random seeds perform well on distinct tasks. However, it remains unclear how an averaged model can achieve strong performance across all tasks.
> > >
> > > **1. It remains unclear how an averaged model can achieve strong performance across all tasks**:
> > >
> > > Prior studies [1,2,3] have shown that models trained from similar initializations tend to maintain a low-loss path between them—a property known as ** linear connectivity**. Linear connectivity describes **the linear relationship between changes in model weights and their corresponding differences in output features resulting from fine-tuning**: the features produced by the weight-interpolated model closely approximate the linear interpolation of features from the two fine-tuned models. This enables effective parameter interpolation, allowing us to leverage multiple models for robust performance.
> > >
> > > For  linear connectivity, we can formally define it as:
> > > $f(\theta_{0}+\alpha \tau_{m})=f(\theta_{0}+\frac{\alpha}{n}\sum_{i=1}^{n}\tau_{i}) \approx f(\theta_{0}) + \frac{\alpha}{n} \sum_{i=1}^{n}\Delta f(\theta_{0}+\tau_{i})=f(\theta_{0})+\alpha \cdot \frac{\Delta f(\theta_{0}+\tau_{1}) + \Delta f(\theta_{0}+\tau_{i}) + \cdots \Delta f(\theta_{0} + \tau_{n}))}{n}$, where $\theta_{0}$ represents the pre-trained model, $\tau_{i}$ is the i-th update weights of LoRA, $\alpha$ is a scale parameter, and $\Delta f(\theta_{0} + \tau_{i}) = f(\theta_{0}+\tau_{i})-f(\theta_{0})$.
> > >
> > > This equation demonstrates that **averaging in weight space is equivalent to averaging in output (feature) space**. This insight explains **why model averaging can effectively combine information from different models and achieve strong performance across diverse sub-tasks**.
> > >
> > > **2. Technical Novelty & Contributions**: While our technical contributions may appear straightforward, their significance lies in several key aspects:
> > >
> > > - Our method explores a fundamentally different direction compared to traditional model averaging. While weight averaging methods primarily focus on **interpolation** between model weights, our approach investigates the potential of **extrapolation** in model fusion, venturing beyond the conventional boundaries defined by existing models. Specifically, interpolation in model merging occurs when the sum of importance weights equals 1 (each model's importance weight is $\frac{1}{n}$ in model averaging), whereas extrapolation allows this sum to exceed 1, enabling exploration beyond the conventional parameter space.
> > >
> > > - We demonstrate that model merging can significantly enhance LoRA performance and effectively **bridge the LoRA-FFT performance gap**. This finding represents **a valuable contribution to the field and warrants broader awareness to stimulate further research in this direction**.
> > >
> > > - Through comprehensive empirical analysis, we identify and characterize **the unique properties of same-task model merging compared to multi-task merging**.
> > >
> > > - Our method achieves competitive performance **while requiring only 2-3 models for merging**, in contrast to previous approaches like model soup that often need 10+ models, making our solution **highly practical**.
> > >
> > >
> > > [1] On the Emergence of Cross-Task Linearity in Pretraining-Finetuning Paradigm. ICML 2024.
> > >
> > > [2] Task Arithmetic in the Tangent Space: Improved Editing of Pre-Trained Models. MeurIPS 2023.
> > >
> > > [3] Going Beyond Linear Mode Connectivity: The Layerwise Linear Feature Connectivity. NeurIPS 2023.

---

### Official Review · Reviewer_6xoV · 2024-11-05

**Soundness:** 2
**Presentation:** 3
**Contribution:** 3
**Rating:** 6
**Confidence:** 3

**Summary:**

This paper proposes a new LoRA-based PEFT method for LLM. It notice that training a LoRA model based on a LLM by initializing with different seeds will result in different performance, which inspires the authors to propose a so-called seedLoRA method that merges various randomly initialized LoRA models to achieve a much better fusion model than any single LoRA model. Experimental results on code, math, and general tasks demonstrate the effectiveness of the proposed seedLoRA. Further more, the LoRA model even outperforms the full-tuning model in some subtasks.

**Strengths:**

1. The idea is interesting. It achieves much better performance than any single LoRA models without introducing any extra computation cost during inference.
2. This paper spends large space analyzing its motivation in Sec. 3.1- Sec. 3.3.
3. The proposed seedLoRA method is straightforward and the performence improvements are not trivial, showing significant improvements over other SOTA methods.
4.  Sec. 4.5 further analyzes the training cost, showing performance increase even in the same training cost compared with the traditional LoRA training.

**Weaknesses:**

1. The analysis of LoRA and full fine-tuning in Sec. 3.2 is not persuasive enough. Can only one case demonstrate the statement "different LoRA models, each trained with unique random seeds, tend to excel in distinct subdomains"? This point may be the most important insight of this paper, so more sophisticated and strict experiments or derivations should be delved into.
2. The description of "Step 3: Distribution Matching:" is insufficiently clear. In Eq.(5), I do not completely understand the calculation of $\tau_m$. $\hat\mu$ and $\hat\sigma$ are the mean and standard deviation of the reference model.  But how can we calculate the mean and the deviation of the best-performing individual model (only one model)? This is the most confused point of mine.

**Questions:**

Besides the questions proposed in weakness, several other minor questions are as below:
1. How to calculate the cosine similarity in Figure 2, Sec. 3.3?
2. "Multi-task methods, such as TIES, are designed to address interference between orthogonal models." how to understand the "interference"?

---

> ### Author Response · Authors · 2024-11-24
> **Response to Reviewer 6xoV**
>
> Thanks for your constructive feedback, we carefully address your concerns below.
>
>
> ### **W1:**
> >The analysis of LoRA and full fine-tuning in Sec. 3.2 is not persuasive enough. Can only one case demonstrate the statement "different LoRA models, each trained with unique random seeds, tend to excel in distinct subdomains"? This point may be the most important insight of this paper, so more sophisticated and strict experiments or derivations should be delved into.
>
> Thank you for your constructive feedback regarding the need for more comprehensive evidence to support our claim about LoRA models with different random seeds excelling in distinct subdomains.
>
> To thoroughly validate this observation, we conducted extensive experiments using general instruction tuning across diverse domains: Factual Knowledge (MMLU), Reasoning Ability (GSM8K, BBH), Multilingual Capability (TyDiQA), Coding Skills (HumanEval).
>
> Our analysis encompasses two model architectures (LLaMA2-7b and Mistral-7b-v0.1) and two fine-tuning methods (LoRA and LoRA+), each tested with three different random seeds (11, 42, 202). The results consistently demonstrate domain-specific excellence across different seeds:
>
> 1.  LLaMA2-7b with LoRA (TULU-v2):
>
> -   seed=11 excels in GSM8K (22.5%) and TyDiQA (51.8%)
> -   seed=42 leads in MMLU (50.1%) and BBH (45.7%)
> -   seed=202 shows strength in HumanEval (15.5%)
>
>  Model|Seed| MMLU | GSM8K | BBH | TyDiQA |  HumanEval | Average
> |---------|---------|------|--------|--------|--------|--------|--------|
> LLaMA2-7b, LoRA | seed=11 | 49.2 | **22.5** | 43.3 | **51.8** | 14.9 | 36.4   |
> LLaMA2-7b, LoRA | seed=42  | **50.1** | 20.0 | **45.7** | 50.5 | 14.6 | 36.2 |
> LLaMA2-7b, LoRA | seed=202  | 49.7 | 16.0 | 45.5 | 51.5 | **15.5** | 35.6 |
>
> 2.  Mistral-7b-v0.1 with LoRA (TULU-v2):
>
> -   seed=11 achieves best performance in MMLU (59.4%) and TyDiQA (59.9%)
> -   seed=202 dominates in GSM8K (50.5%) and BBH (58.7%)
> -   seed=42 shows superior performance in HumanEval (35.5%)
>
>  Model|Seed| MMLU | GSM8K | BBH | TyDiQA |  HumanEval | Average
> |---------|---------|------|--------|--------|--------|--------|--------|
> Mistral, LoRA | seed=11 | **59.4** | 46.0 | 55.0 | **59.9** | 33.8 | 50.8  |
> Mistral, LoRA | seed=42  | 58.8 | 44.0 | 56.6 | 59.5 | **35.5** | 50.9 |
> Mistral, LoRA | seed=202  | 58.2 | **50.5** | **58.7** | 59.0 | 31.4 | 51.6 |
>
> To further validate our findings, we replicated these experiments using LoRA+, an advanced variant of LoRA:
>
> 3.  LLaMA2-7b with LoRA+ (TULU-v2):
>
> -   seed=11 leads in HumanEval (17.9%)
> -   seed=42 excels in MMLU (50.3%)
> -   seed=202 shows strength in GSM8K (25.0%), BBH (46.5%), and TyDiQA (53.1%)
>
>  Model|Seed| MMLU | GSM8K | BBH | TyDiQA |  HumanEval | Average
> |---------|---------|------|--------|--------|--------|--------|--------|
> LLaMA2-7b, LoRA+ | seed=11 | 50.1 | 23.0  | 43.4 | 52.1 | **17.9** | 37.3
> LLaMA2-7b, LoRA+ | seed=42 | **50.3** | 24.5 | 45.4 | 50.9 | 17.4 | 37.7
> LLaMA2-7b, LoRA+ | seed=202 |  49.7 | **25.0** | **46.5** | **53.1** | 16.0 | 38.1
>
>
> 4.  Mistral-7b-v0.1 with LoRA+ (TULU-v2):
>
> -   seed=11 maintains strong performance in HumanEval (34.1%)
> -   seed=42 leads in MMLU (61.2%), BBH (59.7%), and TyDiQA (59.7%)
> -   seed=202 excels in GSM8K (47.0%)
>
>  Model|Seed| MMLU | GSM8K | BBH | TyDiQA |  HumanEval | Average
> |---------|---------|------|--------|--------|--------|--------|--------|
> Mistral-7b-v0.1, LoRA+ | seed=11 | 60.8|  45.0 | 59.4 | 58.2 | **34.1** | 51.51 |
> Mistral-7b-v0.1, LoRA+ | seed=42 | **61.2**|  45.5 | **59.7**| **59.7** | 32.2| 51.66 |
> Mistral-7b-v0.1, LoRA+ | seed=202 | 60.5 | **47.0** | 58.6 | 59.1 | 32.0 | 51.4 |
>
> These comprehensive results consistently demonstrate that models fine-tuned with different random seeds develop distinct strengths across various domains. This systematic pattern holds true across different model architectures (LLaMA2-7b vs. Mistral-7b) and fine-tuning methods (LoRA vs. LoRA+), providing robust evidence for our observation. This finding directly motivated our proposed model fusion method, which leverages these complementary strengths to create a more capable combined model.

---

> > ### Author Response · Authors · 2024-11-24
> > **Response to Reviewer 6xoV - Continue**
> >
> > ### **W2:**
> > >The description of "Step 3: Distribution Matching:" is insufficiently clear. In Eq.(5), I do not completely understand the calculation of  $\tau_m$.  $\hat{\mu}$  and  $\hat{\sigma}$  are the mean and standard deviation of the reference model. But how can we calculate the mean and the deviation of the best-performing individual model (only one model)? This is the most confused point of mine.
> >
> > Thank you for raising this important point. Let us clarify the calculation process:
> >
> > In our experiments, we train multiple individual models (using different random seeds) rather than a single model. For example, with 3 individual models, we obtain 3 sets of mean values and variance values. The final $\hat{\mu}$ and $\hat{\sigma}$ are calculated by averaging these statistics across all models.
> >
> > The mention of the 'best-performing individual model' was intended as an illustrative example of how model importance could potentially be weighted (e.g., using validation performance). However, in our actual implementation and experiments presented in this paper, we use equal weighting across all individual models rather than selecting or emphasizing any particular model. In the revised version, we will update Line 290 to explicitly state that our model merging approach treats all constituent models equally, removing any reference to the selection of a best-performing model.
> >
> > ### **Q1:**
> > >How to calculate the cosine similarity in Figure 2, Sec. 3.3?
> >
> > Thank you for this important clarification question. We apologize for not providing explicit details about the cosine similarity calculation in Section 3.3. Let us clarify the calculation method:
> >
> > The cosine similarity between two LoRA models is computed layer-wise as follows:
> >
> > -   For each LoRA model $i$ and layer $j$, we consider its delta weights $\tau_i^{(j)}=\theta_i^{(j)} -\theta_{pre}^{(j)}$ ,
> > -   These layer-specific delta weights represent the difference between the fine-tuned ($\theta_i^{(j)}$) and pre-trained ($\theta_{pre}^{(j)}$) model parameters at each transformer layer (e.g., layer $j$),
> > -   Layer-wise Calculation: For any two LoRA models (model $i$ and $k$) with delta weights $\tau_i^{(j)}$ and $\tau_k^{(j)}$ at layer $j$, we compute their cosine similarity:  $cos\_sim(\tau_i^{(j)}, \tau_k^{(j)})=(\tau_i^{(j)} \cdot \tau_k^{(i)})/(\|\tau_i^{(j)}\| \|\tau_k^{(j)})$
> >
> > ### **Q2:**
> > >"Multi-task methods, such as TIES, are designed to address interference between orthogonal models." how to understand the "interference"?
> >
> > Thank you for bringing up this point about interference. Let us first clearly define what interference means in model merging, and then clarify the important distinction between multi-task and single-task scenarios.
> >
> > Interference in model merging occurs in two main forms, as identified in TIES [1]:
> >
> > 1.  Interference from redundant parameters: When a parameter is influential (important) for one model but redundant (not important) for others, simple averaging can dilute the influential value. For example:
> >     -  Model A: Parameter P = 5.0 (influential)
> >     -  Model B: Parameter P = 0.1 (redundant)
> >     -  After averaging: P = 2.55 (influential value gets weakened)
> > 2.  Interference from sign disagreement: When models learn opposite directions (positive vs negative values) for the same parameter, averaging can weaken both. For example:
> >     -  Model A: Parameter P = +3.0
> >     -  Model B: Parameter P = -2.0
> >     -  After averaging: P = +0.5 (both values get severely weakened)
> >
> > In multi-task merging, both types of interference are common problems because models trained on different tasks learn very different parameter values. However, for single-task merging (like in our work), these interference issues are less severe because:
> >
> > 1.  Models trained on the same task tend to learn similar parameter values and signs
> > 2.  The high cosine similarity between models (~0.3 vs 0) indicates they encode similar knowledge rather than conflicting information
> >
> > [1] TIES-MERGING: Resolving Interference When Merging Models, NeurIPS 2023.

---

### Official Review · Reviewer_Eapc · 2024-11-07

**Soundness:** 2
**Presentation:** 1
**Contribution:** 2
**Rating:** 5
**Confidence:** 5

**Summary:**

This study proposes SeedLoRA, an approach that aims to improve the performance of LoRA-based Parameter-Efficient Fine-Tuning (PEFT) in Large Language Models (LLMs) by merging models trained on the same task with different random seeds. While existing model merging research largely focuses on combining models trained on different tasks, this study shifts the focus to single-task model merging, highlighting its distinction from conventional multi-task model merging approaches.

SeedLoRA performs the model merging process by weighted averaging n delta models trained with different seeds and normalizing the resulting distribution using the mean and standard deviation of a specific reference model to stabilize the shifted distribution.

Through experiments, the study demonstrates that this approach outperforms conventional LoRA-based fine-tuning (LoRA, LoRA+, DoRA) and existing model merging methods (Model Soup, TIES, DARE), albeit limited to specific complex tasks such as code generation and mathematical reasoning.

**Strengths:**

- The motivation to integrate multiple models trained on the same task with different seeds, based on the observation that LoRA fine-tuning can sometimes exhibit significant performance differences depending on the seed, is intriguing. This aligns with established ensemble learning theories and appears to be a reasonable approach.

- From the perspective of model merging, the study introduces a novel aspect of single-task model merging. The authors' discussion of the differences compared to previous multi-task model merging provides valuable insights.

**Weaknesses:**

Presentation
* While the claims are clear and straightforward, making the paper understandable, the overall writing feels unpolished. Numerous issues, ranging from serious errors that hinder comprehension to minor typos, are apparent throughout the paper.
    * Figure 3: Although emphasized in Section 4.5, results for SeedLoRA are missing from the figure, and its placement is far removed from the section where it is discussed.
    * Line 296: The equation for $\hat{\sigma}$ appears to erroneously calculate it using $\mu$, which seems to be a mistake.
    * Line 71: The word “creat” is likely a typo for “create”. Additionally, many other typos and spacing issues were observed.
* Key technical details are missing or unclear. For instance, in Section 3.4, under "Step 2," there is no explanation of how the importance weight $w_i$ is determined.

Lack of Rationale and Novelty in the Technique
* Since the model weights are averaged (or weighted averaged), the overall weight distribution remains bound to the original distribution of the individual models (as is the case with Model Soup), potentially resulting in no significant shift. Moreover, in Step 3 of Section 3.4, the "reference model" is ultimately a simple average of the model weights. It is difficult to accept that such a minor distribution shift (or even no shift at all) would be practically meaningful. A theoretical analysis in the solution space or references supporting the necessity of this subtle distribution shift is needed.
* Without sufficient rationale for the proposed technique, this work may be seen as merely adapting Model Soup (which already includes some results for simple single-task merging case) to the context of LoRA fine-tuning, thereby lacking sufficient technical novelty.

Validation
* Results for full fine-tuning are missing in most tables and figures. This omission makes it difficult to assess whether the gap between LoRA-based fine-tuning and full fine-tuning has been meaningfully reduced or whether marginal improvements are being overemphasized.
* In Figure 1, LoRA appears to outperform full fine-tuning in some cases, depending on the random seed. However, results are only shown for three seeds (11, 42, 202), raising concerns that SeedLoRA’s performance improvements might also be due to luck rather than robust effectiveness.

**Questions:**

All of my questions are related to the points raised in the Weaknesses section.
1. In Step 2 of Section 3.4, could you clarify how the importance weight $w$ is determined?
2. Could you provide a more detailed theoretical rationale for the proposed model distribution matching?
3. To enhance reliability, could you include results using different seeds beyond 11, 42, and 202, as well as results for full fine-tuning?

---

> ### Author Response · Authors · 2024-11-24
> **Response to Reviewer Eapc**
>
> Thanks for your constructive and inspiring feedback, we carefully address your concerns below.
>
> ### **W1:**
> >While the claims are clear and straightforward, making the paper understandable, the overall writing feels unpolished. Numerous issues, ranging from serious errors that hinder comprehension to minor typos, are apparent throughout the paper.
>
> Thank you for this thorough review and valuable feedback regarding the manuscript's writing quality. We have carefully addressed these concerns through a comprehensive revision of the entire manuscript. Specifically:
>
> - Figure 3: We have thoroughly revised Figure 3 by incorporating the SeedLoRA results and optimizing its placement within the paper to enhance readability and flow of the discussion.
> - Line 296: We have corrected the mathematical error in the equation for $\hat{\sigma}$ on Line 296, and we appreciate you bringing this to our attention.
> - Line 71: Thanks for your careful observation and we have fixed the error on Line 71 as noted.
>
> ### **W2 & Q1:**
> >Key technical details are missing or unclear. For instance, in Section 3.4, under "Step 2," there is no explanation of how the importance weight  $w_i$  is determined.
>
> >In Step 2 of Section 3.4, could you clarify how the importance weight  w  is determined?
>
> Thank you for raising this important point regarding the importance weight calculation in Section 3.4.
>
> In our current experiments, we set the importance weight $w_i = 1$ for all models, treating them as equally important since they differ only in their random seeds. While we used uniform weights in this study, we deliberately introduced $w_i$ in Step 2 to maintain generality in our formulation. This allows for future scenarios where different models might warrant different weights (for example, when using validation performance to determine model importance). We will add this clarification to Section 3.4 to improve clarity.
>
> ### **W3 & Q2:**
> >Since the model weights are averaged (or weighted averaged), the overall weight distribution remains bound to the original distribution of the individual models (as is the case with Model Soup), potentially resulting in no significant shift. Moreover, in Step 3 of Section 3.4, the "reference model" is ultimately a simple average of the model weights. It is difficult to accept that such a minor distribution shift (or even no shift at all) would be practically meaningful. A theoretical analysis in the solution space or references supporting the necessity of this subtle distribution shift is needed.
>
> >Could you provide a more detailed theoretical rationale for the proposed model distribution matching?
>
> Thank you for this careful consideration of our distribution matching approach. We would like to clarify several aspects of SeedLoRA that may have been misunderstood:
>
> 1.  SeedLoRA goes beyond simple weight averaging. While the first step involves weighted averaging (Equation 4), a crucial second step performs distribution matching (Equation 5) that actively rescales the merged weights to match target statistical properties. This is fundamentally different from Model Soup, which only performs weight averaging.
> 2.  Through empirical analysis of weight distributions, we observed:
>
> -   LoRA models trained on the same task consistently exhibit Gaussian-like characteristics, with most weights concentrated near zero
> -   After weight averaging (Step 2), there is a significant distribution shift:
>     -   Weights near zero in original models tend to increase in magnitude
>     -   Larger magnitude weights tend to decrease
>     -   This shift disrupts the careful balance of weight distributions learned during training
> -   For approximately Gaussian distributions, mean and standard deviation are sufficient statistics to characterize and correct these shifts
> -   These patterns hold consistently across different random seeds and training runs
>
> 3.  The "reference model" parameters ($\hat{\mu}$ and $\hat{\sigma}$) are not simply averages of model weights, but rather averages of the statistical properties (mean and standard deviation) of individual models. Our formulation:
>
> -   Preserves the relative relationships between weights while adjusting their scale
> -   Maintains numerical stability through standardization before rescaling
> -   Efficiently captures the key statistical properties we observed in our weight distribution analysis
> -   Critically, helps restore the proper weight distribution that was disrupted during averaging
>
> 4. From a theoretical perspective, maintaining proper statistical distributions is crucial for their performance. Our distribution matching provides:
>
> -   A reliable way to preserve essential statistical properties while avoiding potential instabilities
> -   Robustness through standardization and rescaling steps
> -   Stability by focusing on key statistical moments rather than more complex distribution metrics
> -   A mechanism to correct the significant distribution shifts caused by weight averaging

---

> > ### Author Response · Authors · 2024-11-24
> > **Response to Reviewer Eapc - Continue**
> >
> > ### **W4:**
> > >Without sufficient rationale for the proposed technique, this work may be seen as merely adapting Model Soup (which already includes some results for simple single-task merging case) to the context of LoRA fine-tuning, thereby lacking sufficient technical novelty.
> >
> > Thank you for your constructive comment. While our work builds upon insights from Model Soup, our contributions extend significantly beyond simply adapting it to LoRA fine-tuning:
> >
> > 1.  First to Bridge the Critical LoRA-FFT Performance Gap:
> >
> > -   Narrowing the performance gap between LoRA and Full Fine-Tuning (FFT) is a fundamental challenge in efficient model adaptation.
> > -   To the best of our knowledge, we are the first to demonstrate that model fusion can significantly narrow LoRA-FFT performance gap, a finding that has important implications for the broader ML community.
> > - While Model Soup demonstrates single-task merging, it typically requires merging many models (e.g., 10 or more). In contrast, our SeedLoRA achieves superior performance with just 2-3 models, making it significantly more practical and efficient.
> > -   On GSM8K, SeedLoRA achieves 68.6%, surpassing both full fine-tuning (66.5%) and existing merging methods.
> >
> > 2.  Novel Analysis & Insights:
> >
> > -   Our cosine similarity analysis provides key insights into model relationships: models trained on different tasks exhibit near-zero similarity (orthogonality), while models trained on the same task with different seeds demonstrate high cosine similarity, indicating substantial shared information
> > -   These findings align with weight averaging literature [1,2,3], which demonstrates that under shared pre-training, weights can be linearly interpolated despite architectural non-linearities
> >
> > 3.  Model Complementarity Discovery:
> >
> > -   We demonstrate that LoRA models trained with different seeds develop complementary strengths across task subdomains
> > -   For example, in MMLU, different seeds excel in distinct areas (e.g., professional accounting vs. college mathematics), enabling effective knowledge fusion through merging
> >
> > 4.  Technical Validation:
> >
> > -   Our Weight Distribution Match technique's effectiveness is validated through empirical results and SVD analysis
> > -   The broader non-zero singular value distributions in merged models provide concrete evidence of successful knowledge fusion
> >
> > These contributions collectively advance our understanding of efficient model adaptation. Most importantly, our work resolves a long-standing challenge in LoRA applications by demonstrating that model fusion can achieve performance comparable to or better than full fine-tuning while maintaining LoRA's efficiency benefits.
> >
> >
> > [1] Model soups: averaging weights of multiple fine-tuned models improves accuracy without increasing inference time, ICML 2022.
> >
> > [2] Diverse Weight Averaging for Out-of-Distribution Generalization, NeurIPS 2022.
> >
> > [3] Model Ratatouille: Recycling Diverse Models for Out-of-Distribution Generalization, ICML 2023.

---

> > > ### Author Response · Authors · 2024-11-24
> > > **Response to Reviewer Eapc - Continue**
> > >
> > > ### **W5:**
> > > >Results for full fine-tuning are missing in most tables and figures. This omission makes it difficult to assess whether the gap between LoRA-based fine-tuning and full fine-tuning has been meaningfully reduced or whether marginal improvements are being overemphasized.
> > >
> > > Thank you for this valuable feedback regarding the comparison with full fine-tuning. We have now added comprehensive full fine-tuning results across all experiments (Tables 2-4), which demonstrate that SeedLoRA achieves performance on par with full fine-tuning across diverse tasks:
> > >
> > > 1.  On MetaMathQA (Table 2), SeedLoRA (LoRA) achieves 43.0% average accuracy on GSM8K and MATH combined, effectively matching full fine-tuning (43.2%). Notably, our SeedLoRA (LoRA+) variant slightly outperforms full fine-tuning at 43.6%.
> > >
> > > Method | GSM8K | MATH | Average |
> > > |---------|---------|------|--------|
> > > Full Fine-Tuning | 66.5  | 19.8 |  43.2
> > > LoRA | 64.0 | 15.2 | 39.6
> > > SeedLoRA (LoRA) | 68.6 | 17.3 | 43.0 ($\uparrow$ 3.4)
> > > LoRA+ | 64.8 | 15.6 | 40.2
> > > SeedLoRA (LoRA+) | 69.8 | 17.4 | 43.6 ($\uparrow$ 3.4) |
> > > DoRA | 64.7 | 15.2 | 40.0
> > > SeedLoRA (DoRA) | 68.5 | 17.8 | 43.2 ($\uparrow$ 3.2) |
> > >
> > >
> > > 2. On CodeFeedback (Table 3), SeedLoRA (LoRA) reaches 38.4% average performance across HumanEval and HumanEval+, nearly identical to full fine-tuning's 38.7%. This demonstrates SeedLoRA's effectiveness on complex coding tasks while maintaining the efficiency benefits of parameter-efficient fine-tuning.
> > >
> > > Method | HumanEval | HumanEval+ | Average |
> > > |---------|---------|------|--------|
> > > Full Fine-Tuning | 40.3 |  37.1 | **38.7**
> > > LoRA | 33.5 | 29.1 | 31.3
> > > SeedLoRA (LoRA) | 40.2 | 36.6 | **38.4**
> > > LoRA+ | 34.8 | 31.7 | 33.3
> > > SeedLoRA (LoRA+) | 39.0 | 36.6 | **37.8** |
> > > DoRA | 33.3 | 30.5 | 31.9
> > > SeedLoRA (DoRA) | 37.2 | 32.3 | **34.8** |
> > >
> > > 3.  On TULU-v2 (Table 4), which spans diverse tasks including MMLU, GSM8K, BBH, TyDiQA, and HumanEval, SeedLoRA (LoRA+) achieves 38.8%, approaching full fine-tuning's 39.0%. This comprehensive evaluation across task types reinforces SeedLoRA's ability to match full fine-tuning performance.
> > >
> > > Method | LoRA | SeedLoRA (LoRA) | LoRA+ | SeedLoRA (LoRA+) | DoRA | SeedLoRA (DoRA) | Full Fine-Tuning
> > > |---------|---------|------|--------|--------|--------|--------|--------|
> > > Result | 36.4 | **37.0** | 38.1 | **38.8** | 37.5 | 38.7 | **39.0**
> > >
> > > These results consistently demonstrate that our method can achieve comparable or better performance compared to full fine-tuning while maintaining the computational efficiency of parameter-efficient methods.
> > >
> > > ### **W6 & Q3:**
> > > >In Figure 1, LoRA appears to outperform full fine-tuning in some cases, depending on the random seed. However, results are only shown for three seeds (11, 42, 202), raising concerns that SeedLoRA’s performance improvements might also be due to luck rather than robust effectiveness.
> > >
> > > >To enhance reliability, could you include results using different seeds beyond 11, 42, and 202, as well as results for full fine-tuning?
> > >
> > > Thank you for your valuable suggestion regarding experimental validation. We have conducted additional experiments using three new random seeds (13, 87, 100) on MetaMathQA, complementing our original experiments (seeds 11, 42, 202). The comprehensive results are presented in the table below:
> > >
> > > Method| Task | seed=11 | seed=42 | seed=202 | seed=13 | seed=87 | seed=100 | M(11,42,202) | M(13, 87, 100)| Full FT
> > > |---------|---------|------|--------|--------|--------|--------|--------|------|----|-------------------|
> > > LLaMA-7b, LoRA | GSM8K | 64.0 | 63.8 | 64.1 | 64.2 | 64.7 | 63.5 | 68.6 | 68.9  |  66.5  |
> > > LLaMA-7b, LoRA | MATH  | 15.3 | 15.3 | 14.9 | 15.3 | 15.4 | 14.7 | 17.3 | 17.3  |  19.8   |
> > > LLaMA-7b, LoRA | (Average)  | 39.7 | 39.6 | 39.5 | 39.8 | 40.1 | 39.1 | **43.0** |  **43.1**   |    **43.2**   |
> > >
> > > Where M(11,42,202) represents our originally submitted results using models merged from seeds 11, 42, and 202, while M(13,87,100) represents results from merging models trained with the new seeds 13, 87, and 100.
> > >
> > > The results demonstrate that:
> > >
> > > 1.  The new merged model M(13,87,100) achieves comparable performance to our original M(11,42,202) model (43.1 vs 43.0), providing additional evidence for the robustness of SeedLoRA.
> > > 2.  Both merged models perform competitively with full fine-tuning (43.0/43.1 vs 43.2), supporting our method's effectiveness while being more parameter-efficient.

---

> ### Comment · Reviewer_Eapc · 2024-11-25
>
> I have carefully reviewed all of the author's responses and consolidated my feedback into a comprehensive comment.
>
> The additional results provided by the authors, including full fine-tuning performance and performance with different random seeds, convincingly demonstrate the effectiveness of SeedLoRA. The evidence presented makes it difficult to refute its efficacy.
>
> However, I would like to request further clarification on the following points:
>
> 1. Although I have read the author's responses to Reviewer tN6o, questions regarding “Distribution Matching” and the “Reference Model” remain unresolved.
> - Assuming that each delta model $\tau_i$ is independent and the importance weights are $w_i=1$ (i.e., $\tau_m = \frac{1}{n} \sum_{i=1}^n \tau_i$), based on the information provided in Lines 286–291, it follows that:
> $\mu(\tau_m) = \mu\left(\frac{1}{n} \sum_{i=1}^n \tau_i\right) = \frac{1}{n} \sum_{i=1}^n \mu(\tau_i) = \hat{\mu}.$
> This indicates that the mean of the merged model remains completely identical before and after distribution matching.
> - Similarly, the standard deviation of the merged model is: $\sigma(\tau_m) = \sqrt{\text{Var}(\tau_m)} = \sqrt{\frac{1}{n^2} \sum_{i=1}^n \sigma^2(\tau_i)} = \frac{1}{n} \sqrt{\sum_{i=1}^n \sigma^2(\tau_i)}.$ And, the standard deviation of the reference model, considering the author's intent and correcting any errors in the paper, is $\hat{\sigma} = \frac{1}{n} \sum_{i=1}^n \sigma(\tau_i).$
> - In summary, the distribution matching proposed in the paper, under the general case ($w_i=1$ and the reference model being determined by the statistics of all individual delta models), fails to alter the mean at all but adjusts the standard deviation from $\frac{1}{n} \sqrt{\sum_{i=1}^n \sigma^2(\tau_i)}$ to $\frac{1}{n} \sum_{i=1}^n \sigma(\tau_i).$ While the empirical efficacy of this approach has been demonstrated, a mathematical rationale supporting this technique is necessary. Should this be viewed as merely a heuristic technique? Additionally, please clarify exactly how the reference model was defined in the experiments presented in the paper.
>
> 2. Are all the full fine-tuning results presented in the paper taken from existing literature, or were they obtained through the authors' own training? Additionally, can the authors provide details on the absolute GPU hours required and the GPU models/numbers used to complete the full fine-tuning and LoRA fine-tuning experiments under the experimental settings described in the paper?

---

> ### Author Response · Authors · 2024-11-26
> **Response to Reviewer Eapc**
>
> Thanks for your constructive and inspiring feedback, we carefully address your concerns below.
>
> >1. Although I have read the author's responses to Reviewer tN6o, questions regarding “Distribution Matching” and the “Reference Model” remain unresolved.
>
> Thank you very much for taking the time to read our rebuttal and the other reviewers' comments. We will address your concerns regarding Distribution Matching and the Reference Model.
>
> Firstly, we appreciate your thorough analysis, which is indeed correct.
>
> Regarding your observation about the mean values: You are correct that the mean of the merged model remains identical before and after distribution matching. This is intentional, as the mean values of open-source pre-trained models and fine-tuned models typically hover close to 0, and we aim to preserve this characteristic.
>
> Concerning the standard deviation analysis: Our method adjusts the standard deviation from $\frac{1}{n}\sqrt{\sum_{i=1}^{n} \sigma^{2}(\tau_{i})}$ to $\frac{1}{n}\sum_{i=1}^{n}\sigma(\tau_{i})$. We observed that the averaging merge in step 2 typically results in a smaller standard deviation than individual models ($\frac{1}{n}\sqrt{\sum_{i=1}^{n} \sigma^{2}(\tau_{i})}$ is generally smaller than $\sigma(\tau_{i})$). This reduction in standard deviation occurs because averaging tends to increase weights that were near zero in the original models while decreasing larger magnitude weights. This behavior potentially risks losing important information encoded in the larger magnitude weights of individual models. Therefore, we aim to increase the standard deviation of the averaging model to match $\frac{1}{n}\sum_{i=1}^{n}\sigma(\tau_{i})$, which better represents the standard deviation of individual models.
>
> We can further explain equation (5) from a layer-wise normalization perspective. The equation can be rewritten as: $\tau_{m} = \hat{\sigma} \cdot \frac{\tau_{m}-\mu(\tau_{m})}{\sigma(\tau_{m})} + \hat{u} \approx \hat{\sigma}\cdot\frac{\tau_{m}}{\sigma(\tau_{m})}$, when $\mu(\tau_{m})$ and $\hat{\mu}$ are close to 0. As mentioned earlier, this assumption holds for most pre-trained and fine-tuned models, which we confirmed through analysis of our step 1 fine-tuned models. Given that the mean values are close to 0 and smaller than the standard deviation, we can approximately rewrite the definition of standard deviation as: $\sigma(\tau_{m})\approx\sqrt{[(\tau_{m}^{0}-0)^{2}+(\tau_{m}^{1}-0)^{2} + \cdots +(\tau_{m}^{i}-0)^{2}+\cdots+(\tau_{m}^{N}-0)^{2}]/N}=\sqrt{\sum_{i=1}^{N}(\tau_{m}^{i})^{2}/N}=\frac{||\tau_{m}||}{\sqrt{N}}$. Similarly, $\hat{\sigma}=\frac{1}{n}\sum_{i=1}^{n}\sigma(\tau_{i})=\frac{1}{n}\sum_{i=1}^{n}\frac{||\tau_{i}||}{\sqrt{N}}$
>
> Therefore, equation (5) can be rewritten as: $\tau_{m} = \hat{\sigma} \cdot \frac{\tau_{m}-\mu(\tau_{m})}{\sigma(\tau_{m})} + \hat{u} \approx \hat{\sigma}\cdot\frac{\tau_{m}}{\sigma(\tau_{m})}\approx \frac{1}{n}\sum_{i=1}^{n}||\tau_{i}||\cdot\frac{\tau_{m}}{||\tau_{m}||}$. This reformulation demonstrates that our distribution matching can be interpreted as a layer-wise adaptive weight normalization.
>
>
> >2.  Are all the full fine-tuning results presented in the paper taken from existing literature, or were they obtained through the authors' own training? Additionally, can the authors provide details on the absolute GPU hours required and the GPU models/numbers used to complete the full fine-tuning and LoRA fine-tuning experiments under the experimental settings described in the paper?
>
> Thank you for your questions regarding our experimental details. We confirm that all fine-tuning results presented in the paper are from our own training experiments, not from existing literature. To address your question about computational resources, we have detailed the GPU hours required for both LoRA and Full Fine-tuning approaches on MetaMathQA in the table below. All experiments were conducted on H100-80GB GPUs, maintaining consistent settings including global batch size and gradient accumulation across configurations.
>
> Method | Number of GPU=1, H100-80GB | Number of GPU =4, H100-80GB |
> |---------|---------|------------|
> LLaMA2-7b + LoRA | 17h 19min | 4h 57min |
> LLaMA2-7b + Full Fine-Tuning | / | 5h 47min |

---

> > ### Comment · Reviewer_Eapc · 2024-11-27
> >
> > Thank you for your detailed response.
> >
> > Unfortunately, the response still fails to address the core point regarding the rescaling of the standard deviation. Even without the assumption that the mean is close to 0, the rationale behind the "normalization" mentioned by the authors is already self-evident. The key question is why the standard deviation is rescaled to $\frac{1}{n}\sum_{i=1}^{n}||\tau_{i}||$, and I was unable to find any insight into this in the response.
> >
> > That said, as I mentioned in my previous comments, the effectiveness of SeedLoRA has been validated. Therefore, I will raise my score by one level and also adjust my confidence level upward by one step.

---

> ### Author Response · Authors · 2024-12-03
> **Response to Reviewer Eapc**
>
> Thanks for your constructive feedback, we carefully address your concerns below.
>
> >Even without the assumption that the mean is close to 0, the rationale behind the "normalization" mentioned by the authors is already self-evident. The key question is why the standard deviation is rescaled to $\frac{1}{n}\sum_{i=1}^{n}\tau_{i}$, and I was unable to find any insight into this in the response.
>
> To address your concern, we try to explain it from the perspective of  linear connectivity:
>
> Prior studies [1,2,3] have shown that models trained from similar initializations tend to maintain a low-loss path between them—a property known as linear connectivity. Linear connectivity describes **the linear relationship between changes in model weights and their corresponding differences in output features resulting from fine-tuning**: the features produced by the weight-interpolated model closely approximate the linear interpolation of features from the two fine-tuned models.  We can define  Linear connectivity as: $f(\theta_{0}+\alpha \tau)=f(\theta_{0}) + \alpha \Delta f(\theta_{0}+ \tau)$ and
> $f(\theta_{0}+\alpha \tau_{m})=f(\theta_{0}+\frac{\alpha}{n}\sum_{i=1}^{n}\tau_{i}) \approx f(\theta_{0}) + \frac{\alpha}{n} \sum_{i=1}^{n}\Delta f(\theta_{0}+\tau_{i})=f(\theta_{0})+\alpha \cdot \frac{\Delta f(\theta_{0}+\tau_{1}) + \cdots + \Delta f(\theta_{0} +\tau_{i}) + \cdots + \Delta f(\theta_{0} + \tau_{n}))}{n}$, where $\theta_{0}$ represents the pre-trained model, $\tau_{i}$ is the i-th delta model weights $\tau_{i}=\theta_{i}-\theta_{0}$, $\alpha$ is a scale parameter, and $\Delta f(\theta_{0} + \tau_{i}) = f(\theta_{0}+\tau_{i})-f(\theta_{0})$.
>
> These equations reveal that **averaging in weight space can be interpreted as averaging in output (feature) space**. This insight explains why model averaging effectively combines information from different models and performs well across diverse sub-tasks.
>
> As previously analyzed, distribution matching can be viewed as multiplying by a scale parameter $\frac{\frac{1}{n}\sum_{i=1}^{n}|\tau_{i}|}{|\tau_{m}|}$, which is greater than 1. Therefore, we set $\alpha=\frac{\frac{1}{n}\sum_{i=1}^{n}|\tau_{i}|}{|\tau_{m}|}$ in the above equation for distribution matching. Based on the equation $f(\theta_{0}+\frac{\alpha}{n}\sum_{i=1}^{n}\tau_{i})=f(\theta_{0}+\alpha \tau_{m})=f(\theta_{0})+\alpha \cdot \frac{\Delta f(\theta_{0}+\tau_{1}) + \cdots + \Delta f(\theta_{0}+\tau_{i}) + \cdots + \Delta f(\theta_{0} + \tau_{n}))}{n}$, we can identify two key components in the final outputs: the pre-trained model output $f(\theta_{0})$ and the average of delta outputs $\alpha \cdot \frac{\Delta f(\theta_{0}+\tau_{1}) + \Delta f(\theta_{0}+\tau_{i}) + \cdots \Delta f(\theta_{0} + \tau_{n}))}{n}$. The second component enables the combination of outputs from different models, explaining the effectiveness of model averaging. However, **as the number of models increases, the delta output distribution becomes smoother, potentially diminishing the important information from individual fine-tuned models while increasing the dominance of pre-trained outputs**. To counter this undesirable effect, we introduce a scale hyperparameter $\alpha$ (larger than 1) on outputs to **preserve the valuable information from individual models**.
>
> In SeedLoRA, for the equation $f(\theta_{0}+\frac{\alpha}{n}\sum_{i=1}^{n}\tau_{i})=f(\theta_{0}+\alpha \tau_{m})=f(\theta_{0}+\frac{\frac{1}{n}\sum_{i=1}^{n}||\tau_{i}||}{||\tau_{m}||}\tau_{m})$, we employ the scale value $\alpha=\frac{\frac{1}{n}\sum_{i=1}^{n}||\tau_{i}||}{||\tau_{m}||}$ to ensure the merged delta model **maintains a norm similar to individual delta models**. This approach effectively scales the output features while maintaining stability in a controllable manner. Consequently, **it reinforces the important information from individual models and demonstrates strong performance across various sub-tasks**.
>
> [1] On the Emergence of Cross-Task Linearity in Pretraining-Finetuning Paradigm. ICML 2024.
>
> [2] Task Arithmetic in the Tangent Space: Improved Editing of Pre-Trained Models. MeurIPS 2023.
>
> [3] Going Beyond Linear Mode Connectivity: The Layerwise Linear Feature Connectivity. NeurIPS 2023.

---

### Meta-Review · Area_Chair_X64w · 2024-12-21

**Metareview:**

The work proposes SeedLoRA, a fusion-based approach to improve LoRA fine-tuning by merging models trained with different random seeds. While reviewers acknowledged the potential of single-task model merging and the empirical results demonstrating improvements over traditional LoRA and full fine-tuning in certain tasks, significant concerns remain. Specifically, reviewers Eapc and tN6o highlighted a lack of technical novelty, with the approach appearing to be a straightforward adaptation of model averaging techniques like Model Soup, and insufficient insight into why the proposed distribution matching works effectively. Despite the authors’ efforts during the rebuttal period to clarify methodology and provide additional results, the concerns regarding the underlying rationale and limited innovation still remained.

**Additional Comments On Reviewer Discussion:**

All reviewers have raised concerns about the technical novelty of SeedLoRA, particularly its reliance on model averaging and lack of clear theoretical insights into the effectiveness of the proposed distribution matching approach. They also requested clarifications on key methodological details, including the role of the reference model and the implications of merging models with different random seeds. The authors responded by providing additional results, clarifying computational details, and offering theoretical justifications based on linear connectivity and weight distribution properties. However, while these responses improved clarity and demonstrated empirical effectiveness, the core issues regarding limited innovation and insufficient insight into why the method works remained unresolved. One reviewer has raised the score from 3 to 5, but the overall scores are still insufficient for acceptance.

---

### Decision · Program_Chairs · 2025-01-22

Reject